# Retraining with Predicted Hard Labels Provably Increases Model Accuracy

**Rudrajit Das** [* 1]   **Inderjit S Dhillon** [2 3]   **Alessandro Epasto** [1]   **Adel Javanmard** [1 4]   **Jieming Mao** [1]   **Vahab Mirrokni** [1]
**Sujay Sanghavi** [3]   **Peilin Zhong** [1]

## Abstract

The performance of a model trained with noisy labels is often improved by simply *retraining* the model with its *own predicted hard labels* (i.e., 1/0 labels). Yet, a detailed theoretical characterization of this phenomenon is lacking. In this paper, we theoretically analyze retraining in a linearly separable binary classification setting with randomly corrupted labels given to us and prove that retraining can improve the population accuracy obtained by initially training with the given (noisy) labels. To the best of our knowledge, this is the first such theoretical result. Retraining finds application in improving training with local label differential privacy (DP), which involves training with noisy labels. We empirically show that retraining selectively on the samples for which the predicted label matches the given label significantly improves label DP training at no extra privacy cost; we call this consensus-based retraining. For example, when training ResNet-18 on CIFAR-100 with $\epsilon = 3$ label DP, we obtain more than $6\%$ improvement in accuracy with consensus-based retraining.

## 1 Introduction

We study the simple idea of **retraining** an *already trained* model with its **own predicted hard labels** (i.e., 1/0 labels and *not* the raw probabilities) when the given labels with which the model is initially trained are **noisy**. This is a simple yet effective way to boost a model's performance in the presence of noisy labels. More formally, suppose we train a

discriminative model $\mathcal{M}$ (for a classification problem) on a dataset of $n$ samples and *noisy* label pairs $\{(\boldsymbol{x}_j, \widehat{y}_j)\}_{j=1}^n$. Let $\widehat{\boldsymbol{\theta}}_0$ be the final learned weight/checkpoint of $\mathcal{M}$ and let $\widetilde{y}_j = \mathcal{M}(\widehat{\boldsymbol{\theta}}_0, \boldsymbol{x}_j)$ be the current checkpoint's predicted *hard* label for sample $\boldsymbol{x}_j$. Now, we propose to *retrain* $\mathcal{M}$ with the $\widetilde{y}_j$'s in one of the following two ways:

$(i)$ **Full retraining:** Retrain $\mathcal{M}$ with $\{(\boldsymbol{x}_j, \widetilde{y}_j)\}_{j=1}^n$, i.e., retrain $\mathcal{M}$ with the *predicted* labels of *all* the samples.

$(ii)$ **Consensus-based retraining:** Define $\mathcal{S}_{\text{cons}} := \{j \in \{1, \ldots, n\} \mid \widetilde{y}_j = \widehat{y}_j\}$ to be the set of samples for which the predicted label matches the given noisy label; we call this the *consensus set*. Retrain $\mathcal{M}$ with $\{(\boldsymbol{x}_j, \widetilde{y}_j)\}_{j \in \mathcal{S}_{\text{cons}}}$, i.e., retrain $\mathcal{M}$ with the *predicted* labels of *only the consensus set*.

Intuitively, retraining with predicted hard labels can be beneficial when the underlying classes are "**well-separated**". In such a case, the model can potentially correctly predict the labels of many samples in the training set far away from the decision boundary which were originally incorrectly labeled and presented to it. As a result, the model's accuracy (w.r.t. the actual labels) on the training data can be *significantly higher* than the accuracy of the noisy labels presented to it. Hence, retraining with predicted labels can potentially improve the model's performance. This intuition is illustrated in Figure 1 where we consider a *separable* binary classification problem with noisy labels. The exact details are in Appendix A but importantly, Figures 1a and 1b correspond to versions of this problem with "large" and "small" separation, respectively. Please see the figure caption for detailed discussion but in summary, Figure 1 shows us that *the success of retraining depends on the degree of separation between the classes.*

The motivation for *consensus-based retraining* is that matching the predicted and given labels can potentially yield a smaller but *much more accurate* subset compared to the entire set; such a filtering effect can further improve the model's performance. As we show in Section 5 (see Tables 3 and 11), this intuition bears out in practice.

There are plenty of ideas revolving around training a model with its own predictions, the two most common ones being

---
*Work done while Rudrajit was a PhD student at the University of Texas at Austin and student researcher at Google Research. [1]Google Research [2]Google [3]University of Texas at Austin [4]University of Southern California. Correspondence to: Rudrajit Das <dasrudrajit@google.com>, Alessandro Epasto <aepasto@google.com>, Adel Javanmard <ajavanma@usc.edu>.

*Proceedings of the $42^{nd}$ International Conference on Machine Learning*, Vancouver, Canada. PMLR 267, 2025. Copyright 2025 by the author(s).

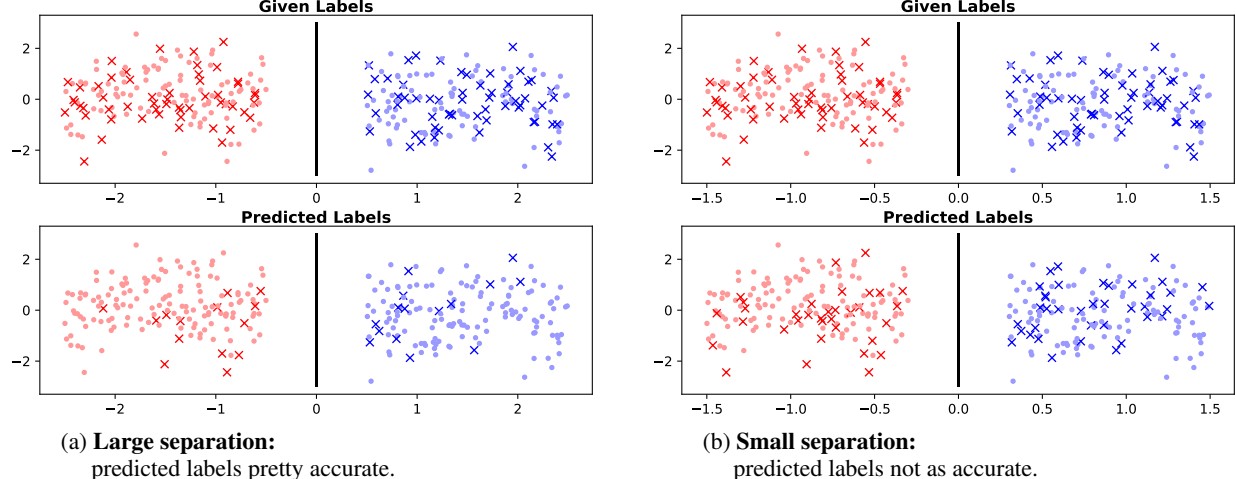

(a) **Large separation:**
predicted labels pretty accurate.

(b) **Small separation:**
predicted labels not as accurate.

*Figure 1.* **Retraining Intuition.** Samples to the right (respectively, left) of the separator (black vertical line in the middle) and colored blue (respectively, red) have *actual* label +1 (respectively, −1). For both classes, the incorrectly labeled samples are marked by crosses (×), whereas the correctly labeled samples are marked by dots (◦) of the appropriate color. The amount of label noise and the number of training samples are the *same* in 1a and 1b. The top and bottom plots show the joint scatter plot of the training samples with the (noisy) labels given to us and the labels predicted by the model after training with the given labels, respectively. Notice that in 1a, *the model correctly predicts the labels of several samples that were given to it with the wrong label – especially, those that are far away from the separator*. This is not quite the case in 1b. This difference gets reflected in the *performance on the test set* after retraining. Specifically, in 1a, *retraining increases the test accuracy to* 97.67% *from* 89%. However, retraining yields no improvement in 1b. **So the success of retraining depends on the inter-class separation; in particular, retraining is beneficial when the classes are well-separated.**

*self-training* (Scudder, 1965; Yarowsky, 1995; Lee et al., 2013) and *self-distillation* (Furlanello et al., 2018; Mobahi et al., 2020); we discuss these and important differences from retraining in Section 2. However, from a *theoretical perspective*, we are not aware of any work proving that retraining a model with its predicted *hard* labels can be beneficial in the presence of label noise in any setting. In Section 4, **we derive the first theoretical result** (to our knowledge) showing that full retraining with hard labels *improves model accuracy*.

The primary reason for our interest in retraining is that it turned out to be a simple yet effective way to improve training with local[1] *label differential privacy (DP)* whose goal is to safeguard the privacy of labels in a supervised ML problem by injecting label noise (see Section 3 for a formal definition). Label DP is used in scenarios where only the labels are considered sensitive, e.g., advertising, recommendation systems, etc. (Ghazi et al., 2021). Importantly, *retraining can be applied on top of any label DP training algorithm at no extra privacy cost*. Our main *algorithmic contribution* is empirically demonstrating *the efficacy of consensus-based retraining in improving label DP training* (Section 5). Three things are worth clarifying here. First, as a meta-idea, retraining is not particularly new; however,

its application – especially with consensus-based filtering – as a light-weight way to improve label DP training at no extra privacy cost is new to our knowledge. Second, we are *not* advocating consensus-based retraining as a SOTA general-purpose algorithm for learning with noisy labels. Third, we do not view full retraining to be an algorithmic contribution; we consider it for theoretical analysis and as a baseline for consensus-based retraining.

Our **main contributions** can be summarized as follows:

**(a)** In Section 4, we consider a linearly separable binary classification problem wherein the data (feature) dimension is $d$, and we are given randomly flipped labels with the label flipping probability being $p < \frac{1}{2}$ independently for each sample. *Our main result is proving that full retraining with the predicted hard labels improves the population accuracy obtained by initially training with the given labels*, provided that $p$ is close enough to $\frac{1}{2}$ and the dataset size $n$ satisfies $\frac{d}{(1-2p)^2} \log \frac{d}{(1-2p)^2} \lesssim n \lesssim \frac{d^2}{(1-2p)^2}$ ("$\lesssim$" means bounded asymptotically, ignoring constant factor multiples); see Remark 4.10 for details. In addition, our results show that retraining becomes more beneficial as the amount of label noise (i.e., $p$) increases or as the degree of separation between the classes increases. To the best of our knowledge, **these are the first theoretical results** quantifying the benefits of retraining with predicted hard labels in the presence

---

[1] In this paper, we focus only on local label DP. So throughout the paper, we will mostly omit the word "local" for conciseness.

of label noise. *The analysis of retraining is particularly challenging* because of the dependence of the predicted labels on the entire training set and the non-uniform/sample-dependent nature of label noise in the predicted labels; see the discussion after the statement of Theorem 4.8.

**(b)** In Section 5, we show the promise of **consensus-based retraining** (i.e., retraining on only those samples for which the predicted label matches the given noisy label) as a simple way to improve the performance of any label DP algorithm, at no extra privacy cost. For e.g., when training ResNet-18 on CIFAR-100 with $\epsilon = 3$ label DP, we obtain $6.4\%$ improvement in accuracy with consensus-based retraining (see Table 2). The corresponding improvement for a small BERT model trained on AG News Subset (a news classification dataset) with $\epsilon = 0.5$ label DP is $11.7\%$ (see Table 10).

## 2 Related Work

**Self-training (ST).** Retraining is similar in spirit to ST (Scudder, 1965; Yarowsky, 1995; Lee et al., 2013; Sohn et al., 2020) which is the process of progressively training a model with its own predicted hard labels in the *semi-supervised* setting. Our focus in this work is on the *fully supervised* setting. This is different from ST (in the semi-supervised setting) which typically selects samples based on the model's confidence and hence, we call our algorithmic idea of interest *retraining* to distinguish it from ST. In fact, we show that our consensus-based sample selection strategy leads to better performance than confidence-based sample selection in Appendix J. There is a vast body of work on ST and related ideas; see Amini et al. (2022) for a survey. On the theoretical side also, there are several papers showing and quantifying different kinds of benefits of ST and related ideas (Carmon et al., 2019; Raghunathan et al., 2020; Kumar et al., 2020; Chen et al., 2020; Oymak & Gulcu, 2020; Wei et al., 2020; Zhang et al., 2022). But *none of these works characterize the pros/cons of ST or any related algorithm in the presence of noisy labels*. In contrast, we show that retraining can provably improve accuracy in the presence of label noise in Section 4. Empirically, ST-based ideas have been proposed to improve learning with noisy labels (Reed et al., 2014; Tanaka et al., 2018; Han et al., 2019; Nguyen et al., 2019; Li et al., 2020; Goel et al., 2022); but these works do not have rigorous theory. In the context of theory on label noise and model's confidence, Zheng et al. (2020) show that if the model's predicted score for the observed label is small, then the observed label is likely not equal to the true label. Note that the results of Zheng et al. (2020) pertain to the correctness of the observed labels, whereas our results pertain to the correctness of the predicted labels.

**Self-distillation (SD).** Retraining is also similar in principle to SD (Furlanello et al., 2018; Mobahi et al., 2020), where a teacher model is first trained with provided hard labels and

then its predicted *soft* labels are used to train a student model with the same architecture as the teacher. In contrast, we use predicted *hard* labels in retraining. Also, SD is usually employed with a temperature parameter (Hinton et al., 2015) to force the teacher and student models to be different; we do not have any such parameter in retraining. SD is known to ameliorate learning in the presence of noisy labels (Li et al., 2017) and this has been theoretically analyzed by Dong et al. (2019); Das & Sanghavi (2023). Dong et al. (2019) propose their own SD algorithm that uses *dynamically updated soft* labels and provide some complicated conditions of when their algorithm can learn the correct labels in the presence of noisy labels. In contrast, we analyze retraining with *fixed hard* labels. Das & Sanghavi (2023) analyze the standard SD algorithm in the presence of noisy labels with *fixed soft* labels but their analysis in the classification setting requires some strong assumptions such as access to the population, feature maps of all points in the same class having the same inner product, etc. We do not require such strong assumptions in this paper (in fact, we present sample complexity bounds).

**Label differential privacy (DP).** Label DP (described in detail in Section 3) is a relaxation of full-data DP wherein the privacy of only the labels (and not the features) is safeguarded (Chaudhuri & Hsu, 2011; Beimel et al., 2013; Wang & Xu, 2019; Ghazi et al., 2021; Malek Esmaeili et al., 2021; Ghazi et al., 2022; Badanidiyuru Varadaraja et al., 2023). In this work, we are not trying to propose a SOTA label DP algorithm (with an ingenious noise-injection scheme); instead, we advocate retraining as a simple post-processing step that can be applied on top of any label DP algorithm (regardless of the noise-injection scheme) to improve its performance, at no extra privacy cost. Similar to our goal, Tang et al. (2022) apply techniques from unsupervised and semi-supervised learning to improve label DP training. In particular, one of their steps involves keeping the given noisy label of a sample only if it matches a pseudo-label generated by unsupervised learning. This is similar in spirit to our consensus-based retraining scheme but a crucial difference is that we do not perform any unsupervised learning; we show that matching the given noisy label to the model's own predicted label is itself pretty effective. Further, unlike our work, Tang et al. (2022) do not have any rigorous theory.

## 3 Preliminaries

**Notation.** For two functions $f(n)$ and $g(n)$, we write $f(n) \lesssim g(n)$ if there exists $n_0$ and a constant $C > 0$ such that for all $n \geq n_0$, $f(n) \leq Cg(n)$. For any positive integer $m \geq 1$, we denote the set $\{1, \ldots, m\}$ by $[m]$. Let $e_i$ denote the $i^{\text{th}}$ canonical vector, namely, the vector of all zeros except a one in the $i^{\text{th}}$ coordinate. We denote the $\ell_2$ norm of a vector $v$ by $\|v\|_{\ell_2}$, and the operator norm of a matrix

$M$ by $\|M\|$. The unit $d$-dimensional sphere (i.e., the set of $d$-dimensional vectors with unit norm) is denoted by $\mathbb{S}^{d-1}$. For a random variable $X$, its sub-gaussian norm, denoted by $\|X\|_{\psi_2}$, is defined as $\|X\|_{\psi_2} := \sup_{q \geq 1} q^{-1/2} \left( \mathbb{E}|X|^q \right)^{1/q}$.[2] In addition, for random vector $\boldsymbol{X} \in \mathbb{R}^d$, its sub-gaussian norm is defined as $\|\boldsymbol{X}\|_{\psi_2} = \sup_{\boldsymbol{z} \in \mathbb{S}^{d-1}} \|\langle \boldsymbol{X}, \boldsymbol{z} \rangle\|_{\psi_2}$. We denote the CDF and complementary CDF (CCDF) of a standard normal variable (i.e., distributed as $\mathcal{N}(0,1)$) by $\Phi(.)$ and $\Phi^c(.)$, respectively.

**Definition 3.1** (**Label differential privacy (DP)**). A randomized algorithm $\mathcal{A}$ taking as input a dataset and with range $\mathcal{R}$ is said to be $\epsilon$-labelDP if for any two datasets D and D$'$ differing in the *label* of a single example and for any $\mathcal{S} \subseteq \mathcal{R}$, it holds that $\mathbb{P}\left( \mathcal{A}(\text{D}) \in \mathcal{S} \right) \leq e^\epsilon \mathbb{P}\left( \mathcal{A}(\text{D}') \in \mathcal{S} \right)$.

Local label DP training involves injecting noise into the labels and then training with these noisy labels. The simplest way of injecting label noise to ensure label DP is randomized response (RR) introduced by Warner (1965). Specifically, suppose we require $\epsilon$-labelDP for a problem with $C$ classes, then the distribution of the output $\widehat{y}$ of RR when $y$ is the true label is given by: $\mathbb{P}\left( \widehat{y} = y \right) = \frac{e^\epsilon}{e^\epsilon + C - 1}$, and $\mathbb{P}\left( \widehat{y} = z \right) = \frac{1}{e^\epsilon + C - 1}$ for any $z \neq y$. Our label noise model in Section 4 (Equation (4.2)) is actually RR for 2 classes. Based on vanilla RR, more sophisticated ways to inject label noise for better performance under label DP have been proposed (Ghazi et al., 2021; Malek Esmaeili et al., 2021). Our empirical results in Section 5 are with RR and the method of Ghazi et al. (2021).

# 4 Full Retraining in the Presence of Label Noise: Theoretical Analysis

Here we will analyze full retraining (as introduced in Section 1) for a linear setting with noisy labels. Since full retraining is the only kind of retraining we consider here, we will omit the word "full" subsequently in this section.

**Problem setting:** We consider binary classification in a linearly separable[3] setting similar to the classical Gaussian mixture model. We will first describe the classical Gaussian mixture model and then describe our setting of interest, which is endowed with a positive margin.

In the classical Gaussian mixture model, each data point belongs to one of two classes $\{+1, -1\}$ with corresponding probabilities $\pi_+$, $\pi_-$, such that $\pi_+ + \pi_- = 1$. The feature vector $\boldsymbol{x} \in \mathbb{R}^d$ for a data point with label $y \in \{+1, -1\}$ is drawn independently from $\mathcal{N}(y\boldsymbol{\mu}, \boldsymbol{\Sigma})$, where $\boldsymbol{\mu} \in \mathbb{R}^d$ and $\boldsymbol{\Sigma} \in \mathbb{R}^{d \times d}$. In other words the mean of the feature vector is $\pm \boldsymbol{\mu}$ depending on the data point's label, and $\boldsymbol{\Sigma}$ is the

covariance matrix of the feature vector. We let $\gamma := \|\boldsymbol{\mu}\|_{\ell_2}$.

**Gaussian mixture model with positive margin:** Here each data point $(\boldsymbol{x}, y)$ is generated independently by first sampling the label $y \in \{+1, -1\}$ with probabilities $\pi_+, \pi_-$, and then generating the feature vector $\boldsymbol{x} \in \mathbb{R}^d$ as:

$$\boldsymbol{x} = y(1 + u)\boldsymbol{\mu} + \boldsymbol{\Sigma}^{1/2} \boldsymbol{z}, \quad \text{where} \qquad (4.1)$$

- $u > 0$ is drawn from a sub-gaussian distribution with unit sub-gaussian norm (i.e., $\|u\|_{\psi_2} = 1$).

- $\boldsymbol{z} \sim \mathcal{N}(\boldsymbol{0}, \boldsymbol{I}_d)$ independent of $u$.

- $\boldsymbol{\Sigma} \in \mathbb{R}^{d \times d}$ is the covariance matrix in the space orthogonal to $\boldsymbol{\mu}$. We have $\boldsymbol{\Sigma}^{1/2} \boldsymbol{\mu} = \boldsymbol{0}$ and $\boldsymbol{\Sigma}$ is of rank $d - 1$. Also, $\lambda_{\min} > 0$ and $\lambda_{\max}$ are the minimum non-zero eigenvalue and the maximum eigenvalue of $\boldsymbol{\Sigma}$.

Note that under this data model, the projection of datapoints on the space orthogonal to $\boldsymbol{\mu}$ is Gaussian. Along $\boldsymbol{\mu}$, we have $\langle \boldsymbol{x}, \boldsymbol{\mu} \rangle = y(1 + u) \|\boldsymbol{\mu}\|_{\ell_2}^2$, and so the randomness of data along $\boldsymbol{\mu}$ is due to $u$. Further, since $u > 0$, $\text{sign}(\langle \boldsymbol{x}, \boldsymbol{\mu} \rangle) = y$. *Thus, $\boldsymbol{\mu}$ is a separator for the data w.r.t. the labels.* In addition, we have a margin of at least $\gamma = \|\boldsymbol{\mu}\|_{\ell_2}$; this is because $\frac{y \langle \boldsymbol{x}, \boldsymbol{\mu} \rangle}{\|\boldsymbol{\mu}\|_{\ell_2}} = (1 + u) \|\boldsymbol{\mu}\|_{\ell_2} \geq \gamma$. We will mostly refer to the margin $\gamma$ as the "degree of separation".

We are given access to a training set $\mathcal{T} = \{(\boldsymbol{x}_i, \widehat{y}_i)\}_{i \in [n]}$ where for each $i \in [n]$, $\widehat{y}_i$ is a noisy version of the true label $y_i$ (which we do not observe). Specifically:

$$\widehat{y}_i = \begin{cases} y_i & \text{with probability } 1 - p, \\ -y_i & \text{with probability } p, \end{cases} \qquad (4.2)$$

for some $p < 1/2$, and independently for each $i \in [n]$.[4]

## 4.1 Initial Training

Given the training set $\mathcal{T} = \{(\boldsymbol{x}_i, \widehat{y}_i)\}_{i \in [n]}$, we consider the following linear classifier model[5] (also used in Carmon et al. (2019)):

$$\widehat{\boldsymbol{\theta}}_0 = \frac{1}{n} \sum_{i=1}^n \widehat{y}_i \boldsymbol{x}_i. \qquad (4.3)$$

This classifier's predicted label for a sample $\boldsymbol{x} \in \mathbb{R}^d$ is $\text{sign}(\langle \boldsymbol{x}, \widehat{\boldsymbol{\theta}}_0 \rangle)$. Note that $\langle \boldsymbol{x}, \widehat{\boldsymbol{\theta}}_0 \rangle = \frac{1}{n} \sum_i \widehat{y}_i \langle \boldsymbol{x}_i, \boldsymbol{x} \rangle$ is a

---

[2] Refer to Vershynin (2010) for other equivalent definitions.

[3] More specifically, there exists a linear separator that perfectly classifies the data points according to their labels.

[4] In the context of $\epsilon$-labelDP for a binary classification problem with randomized response, $p = \frac{1}{e^\epsilon + 1}$.

[5] This is a simplification to the least squares' solution (LSS) obtained by excluding the empirical covariance matrix's inverse. The LSS can be analyzed by bounding the deviation of the empirical covariance matrix from the population covariance matrix (which shrinks as $n \to \infty$), then adapting our current analysis to features pre-multiplied by the population covariance matrix's inverse, and accounting for this deviation. This would only make the math more tedious without adding any meaningful insights.

weighted average of the noisy labels in the training set, with the weights being $\frac{1}{n}\langle x_i, x \rangle$. So it is similar to kernel methods with the inner product kernel.

Our next result bounds the probability of $\widehat{\theta}_0$ misclassifying a fixed test point $x$, with respect to the randomness in the training set $\mathcal{T}$.

**Theorem 4.1** (**Initial training**). *Consider $x \notin \mathcal{T}$ and let $y$ be its true label. We have*

$$\mathbb{P}(\text{sign}(\langle x, \widehat{\theta}_0 \rangle) \neq y) \geq \alpha_0(x) :=$$

$$\frac{1}{2\sqrt{2\pi}} \exp\left(-\frac{5(1 + \sqrt{n}(1-2p))^2 \langle x, \mu \rangle^2}{\left\|\Sigma^{1/2} x\right\|_{\ell_2}^2}\right) \text{ and}$$

$$\mathbb{P}(\text{sign}(\langle x, \widehat{\theta}_0 \rangle) \neq y) \leq \widetilde{\alpha}_0(x) :=$$

$$\frac{1}{2} \exp\left(-\frac{n(1-2p)^2 \langle x, \mu \rangle^2}{8\left\|\Sigma^{1/2} x\right\|_{\ell_2}^2}\right) + 2\exp\left(-\frac{n}{32}(1-2p)^2\right).$$

The proof of Theorem 4.1 is in Appendix B. Notice that the learned classifier $\widehat{\theta}_0$ is more likely to be wrong on the samples that are less aligned with (or closer to orthogonal to) the ground truth separator, i.e., $\mu$. We can view $\widehat{\theta}_0$ as a *noisy* label provider (a.k.a. pseudo-labeler) where the degree of label noise is **non-uniform** or **sample-dependent** unlike the original noisy source used to learn $\widehat{\theta}_0$. Specifically, for a sample $x$ with true label $y$ and predicted label $\widetilde{y} = \text{sign}(\langle x, \widehat{\theta}_0 \rangle)$, we have:

$$\widetilde{y} = \begin{cases} y \text{ with probability at least } \geq 1 - \widetilde{\alpha}_0(x) \text{ and} \\ -y \text{ with probability at most } \leq \widetilde{\alpha}_0(x), \end{cases}$$

where $\widetilde{\alpha}_0(x)$ is as defined in Theorem 4.1. We will next define the population error of a classifier $\widehat{\theta}$ as

$$\text{err}(\widehat{\theta}) := \mathbb{E}_{(x,y)}\left[\mathbb{P}(\text{sign}(\langle x, \widehat{\theta} \rangle) \neq y)\right], \qquad (4.4)$$

where the probability (inside the expectation) is w.r.t. the randomness in the training set used to learn $\widehat{\theta}$. Similarly, the population accuracy of $\widehat{\theta}$ is defined as

$$\text{acc}(\widehat{\theta}) := \mathbb{E}_{(x,y)}\left[\mathbb{P}(\text{sign}(\langle x, \widehat{\theta} \rangle) = y)\right] = 1 - \text{err}(\widehat{\theta}). \tag{4.5}$$

Depending on the context, our results and discussions will sometimes be presented in terms of error (lower is better) and at other times in terms of accuracy (higher is better).

Our next result bounds the error of the classifier $\widehat{\theta}_0$.

**Theorem 4.2** (**Initial training's population error**). *We have*

$$\text{err}(\widehat{\theta}_0) \geq \frac{\left(1 - e^{-\frac{d}{16}}\right)}{4\sqrt{2\pi}} \exp\left(-\frac{160(1 + \sqrt{n}(1-2p))^2 \gamma^4}{\lambda_{\min}^2 d}\right),$$
$$(4.6)$$

$$\text{err}(\widehat{\theta}_0) \leq \frac{1}{2} \exp\left(-\frac{n(1-2p)^2 \gamma^4}{16\lambda_{\max}^2 d}\right) + e^{-\frac{d}{8}}$$
$$+ 2\exp\left(-\frac{n}{32}(1-2p)^2\right). \quad (4.7)$$

The proof of Theorem 4.2 is in Appendix C.

*Remark* 4.3 (**Tightness of error bounds**). When $n \lesssim \frac{\lambda_{\max}^2 d^2}{(1-2p)^2 \gamma^4}$, $\gamma \lesssim (\lambda_{\max}^2 d)^{1/4}$ and $\frac{\lambda_{\max}}{\lambda_{\min}} \lesssim 1$, then the lower and upper bounds for $\text{err}(\widehat{\theta}_0)$ in Theorem 4.2 match (up to constant factors).

Based on Theorem 4.2, we have the following corollary.

**Corollary 4.4** (**Initial training's sample complexity**). *For any $\delta > 2e^{-d/8} + 4\exp\left(-\frac{n}{32}(1-2p)^2\right)$, having number of samples $n \geq 8\lambda_{\max}^2 \frac{\log 1/\delta}{(1-2p)^2} \frac{d}{\gamma^4}$ ensures $\text{acc}(\widehat{\theta}_0) > 1 - \delta$. In particular, when the label flipping probability satisfies $p > 2e^{-d/8} + 4\exp\left(-\frac{n}{32}(1-2p)^2\right)$, then $n \geq 8\lambda_{\max}^2 \frac{\log 1/p}{(1-2p)^2} \frac{d}{\gamma^4}$ ensures $\text{acc}(\widehat{\theta}_0) > 1 - p$, i.e., our learned classifier $\widehat{\theta}_0$ has better accuracy than the source providing noisy labels (used to learn $\widehat{\theta}_0$).*

*Remark* 4.5 (**Effect of degree of separation**). As the parameter quantifying the degree of separation $\gamma$ decreases, the error bound in Theorem 4.2 increases and the sample complexity required to outperform the noisy label source in Corollary 4.4 increases. This is consistent with our intuition that a classification task should become harder as the degree of separation reduces; we saw this in Figure 1.

We conclude this subsection by deriving an information-theoretic lower bound on the sample complexity of *any classifier* to argue that $\widehat{\theta}_0$ attains the optimal sample complexity with respect to $d$ and $p$.

**Theorem 4.6** (**Information-theoretic lower bound on sample complexity**). *With a slight generalization of notation, let $\text{acc}(\widehat{\theta}; \mu)$ denote the accuracy of the classifier $\widehat{\theta}$ as per Equation (4.5), when the ground truth separator is $\mu$. We also consider the case of $\Sigma = \mathcal{P}_\mu^\perp$, viz., the projection matrix onto the space orthogonal to $\mu$. For any classifier $\widehat{\theta}$ learned from $\mathcal{T} := \{(x_j, \widehat{y}_j)\}_{j \in [n]}$, in order to achieve $\inf_{\mu \in \mathbb{S}^{d-1}} \text{acc}(\widehat{\theta}; \mu) \geq 1 - \delta$, the condition $n = \Omega\left(\frac{(1-\delta)}{(1-2p)^2}d\right)$ is necessary in our problem setting.*

It is worth mentioning that there is a similar lower bound in Gentile & Helmbold (1998) for a different classification setting. In contrast, Theorem 4.6 is tailored to our setting and moreover, the proof technique is also different and interesting in its own right. Specifically, for the proof of Theorem 4.6, we follow a standard technique in proving minimax lower bounds which is to reduce the problem of interest to an appropriate multi-way hypothesis testing problem; this is accompanied by the application of the conditional version of Fano's inequality and some ideas from high-dimensional

geometry. This proof is in Appendix D. There are also a few other noteworthy lower bounds (Cai & Wei, 2021; Im & Grigas, 2023; Zhu et al., 2024) in more general settings than ours; naturally, their bounds are weaker than Theorem 4.6. We discuss these bounds in Appendix E.

*Remark* 4.7 (**Minimax optimality of sample complexity**). Note that the dependence of the sample complexity on $d$ and $p$ in Corollary 4.4 matches that of the lower bound in Theorem 4.6. Thus, our sample complexity bound in Corollary 4.4 is optimal with respect to $d$ and $p$.

## 4.2 Retraining

We first label the training set using $\widehat{\boldsymbol{\theta}}_0$. Denote by $\widetilde{y}_i = \text{sign}(\langle \boldsymbol{x}_i, \widehat{\boldsymbol{\theta}}_0 \rangle)$ the predicted label for sample $\boldsymbol{x}_i$. We then retrain using these predicted labels. Our learned classifier here is similar to the one in Section 4.1, except that the observed labels are replaced by the predicted labels. Specifically, our retraining classifier model is the following:

$$\widehat{\boldsymbol{\theta}}_1 = \frac{1}{n} \sum_i \widetilde{y}_i \boldsymbol{x}_i \,. \tag{4.8}$$

$\widehat{\boldsymbol{\theta}}_1$'s predicted label for a sample $\boldsymbol{x} \in \mathbb{R}^d$ is $\text{sign}(\langle \boldsymbol{x}, \widehat{\boldsymbol{\theta}}_1 \rangle)$. Our next result bounds the probability of $\widehat{\boldsymbol{\theta}}_1$ misclassifying a fixed test point $\boldsymbol{x}$, with respect to the randomness in the training set $\mathcal{T}$ (similar to Theorem 4.1).

**Theorem 4.8** (**Retraining**). *Suppose that* $\frac{n}{d} > \frac{4\lambda_{\max}}{\gamma^2(1-2p)}$, $nd > \frac{\gamma^4}{\lambda_{\max}^2}$, *and* $d \geq 7$. *Consider* $\boldsymbol{x} \notin \mathcal{T}$ *and let* $y$ *be its true label. We have*

$$\mathbb{P}(\text{sign}(\langle \boldsymbol{x}, \widehat{\boldsymbol{\theta}}_1 \rangle) \neq y) \leq \alpha_1(\boldsymbol{x}), \text{ where}$$

$$\alpha_1(\boldsymbol{x}) \coloneqq 2 \exp\left(-\frac{n(1-2q')^2 \langle \boldsymbol{x}, \boldsymbol{\mu} \rangle^2}{64(\langle \boldsymbol{x}, \boldsymbol{\mu} \rangle^2 + \left\|\boldsymbol{\Sigma}^{1/2}\boldsymbol{x}\right\|_{\ell_2}^2)}\right)$$
$$+ 4 \exp\left(-\frac{n}{32}(1-2p)^2\right)$$
$$+ \frac{n}{2}\left(\exp\left(-\frac{\gamma^4}{8\lambda_{\max}^2}(1-2p')\frac{n}{d}\right) + e^{-d/16}\right)e^{d/n},$$

*with* $q' = \exp\left(-\frac{n(1-2p)\gamma^2}{40\lambda_{\max}}\right)$ *and* $p' = \left(1 + \frac{3\gamma^4}{8\lambda_{\max}^2 nd}\right)p$.

The proof of Theorem 4.8 is in Appendix F.

**Technical challenges:** The proof of Theorem 4.8 is especially challenging due to the following reasons:

**(i)** The source of the predicted labels $\{\widetilde{y}_i\}_{i=1}^n$ used by $\widehat{\boldsymbol{\theta}}_1$, viz., $\widehat{\boldsymbol{\theta}}_0$, is a *non-uniform noisy label provider*, as we discussed after Theorem 4.1.

**(ii)** More importantly, $\widehat{\boldsymbol{\theta}}_0$ depends on all the samples in the training set and hence each predicted label $\widetilde{y}_i$ is dependent

on the entire training set. *This dependence of the predicted labels on all the data points makes the analysis highly challenging because standard techniques under independence are not applicable here.*[6]

The high-level proof idea for Theorem 4.8 is that for each sample $\boldsymbol{x}_\ell = y_\ell(1 + u_\ell)\boldsymbol{\mu} + \boldsymbol{\Sigma}^{1/2}\boldsymbol{z}_\ell$ in the training set, we carefully curate a dummy label $\widetilde{y}'_\ell$ that does not depend on $\{\boldsymbol{z}_k\}_{k \neq \ell}$ but depends on all $\{u_k\}_{k \in [n]}$; see (F.3) in Appendix F. We show that with high probability, the labels predicted by $\widehat{\boldsymbol{\theta}}_0$ on the training set $\{\widetilde{y}_\ell\}_{\ell \in [n]}$ match the corresponding dummy labels $\{\widetilde{y}'_\ell\}_{\ell \in [n]}$. We analyze the misclassification error of $\widehat{\boldsymbol{\theta}}_1$ under this high probability "good" event with the dummy labels $\{\widetilde{y}'_\ell\}_{\ell \in [n]}$, which are more conducive to analysis (because they do not depend on $\{\boldsymbol{z}_k\}_{k \neq \ell}$), and also bound the probability of the "bad" event (i.e., $\exists$ at least one $\ell$ such that $\widetilde{y}_\ell \neq \widetilde{y}'_\ell$). We provide an outline of these steps in detail at the beginning of Appendix F.

We will now provide an upper bound on the error (defined in (4.4)) of $\widehat{\boldsymbol{\theta}}_1$.

**Theorem 4.9** (**Retraining's population error**). *Suppose the conditions of Theorem 4.8 hold. Then, we have*

$$\text{err}(\widehat{\boldsymbol{\theta}}_1) \leq 2 \exp\left(-\frac{n(1-2q')^2\gamma^4}{64(\gamma^4 + 2\lambda_{\max}^2 d)}\right)$$
$$+ 2e^{-d/8} + 4\exp\left(-\frac{n}{32}(1-2p)^2\right)$$
$$+ \frac{n}{2}\left(\exp\left(-\frac{\gamma^4}{8\lambda_{\max}^2}(1-2p')\frac{n}{d}\right) + e^{-d/16}\right)e^{d/n}, \tag{4.9}$$

*with* $q' = \exp\left(-\frac{n(1-2p)\gamma^2}{40\lambda_{\max}}\right)$ *and* $p' = \left(1 + \frac{3\gamma^4}{8\lambda_{\max}^2 nd}\right)p$.

The proof of Theorem 4.9 is in Appendix G. As expected, (4.9) is decreasing in $\gamma$, because the classification task becomes easier as the degree of separation increases.

**Retraining vs. Initial training.** We will now compare the error bounds of the initial training model $\widehat{\boldsymbol{\theta}}_0$ (4.6) from Theorem 4.2 with that of the retraining model $\widehat{\boldsymbol{\theta}}_1$ (4.9) from Theorem 4.9. Note that the dominant term in the upper bound on $\text{err}(\widehat{\boldsymbol{\theta}}_1)$ is the last term in (4.9) which has $(1-2p')$ inside the exponent, and $p'$ approaches $p$ as $n$ and $d$ grow. In contrast, the lower bound on $\text{err}(\widehat{\boldsymbol{\theta}}_0)$ in (4.6) has $(1 - 2p)^2$ inside the exponent. *This is the main observation that indicates retraining can improve model accuracy, and this improvement becomes increasingly significant as $p \to \frac{1}{2}$.* We will now formalize this observation by providing sufficient conditions under which $\text{err}(\widehat{\boldsymbol{\theta}}_1) < \text{err}(\widehat{\boldsymbol{\theta}}_0)$ or equivalently, $\text{acc}(\widehat{\boldsymbol{\theta}}_1) > \text{acc}(\widehat{\boldsymbol{\theta}}_0)$.

*Remark* 4.10 (**When does retraining improve accuracy?**). We consider an asymptotic regime where $n, d \to \infty$. Also,

---

[6] The analysis of $\widehat{\boldsymbol{\theta}}_0$ (Theorem 4.1) is relatively simpler because the given noisy labels $\{\widehat{y}_i\}_{i=1}^n$ are independent across samples.

suppose that $\frac{\lambda_{\max}}{\lambda_{\min}} \lesssim 1$. By comparing (4.6) with (4.9), we obtain the following sufficient conditions. There exists an absolute constant $c_0$ such that if $p \in \left( \frac{1}{2} - c_0, \frac{1}{2} \right)$ and

$$\frac{\lambda_{\min}^2 d}{\gamma^4 (1-2p)^2} \log\left( \frac{\lambda_{\min}^2 d}{\gamma^4 (1-2p)^2} \right) \lesssim n \lesssim \frac{\lambda_{\min}^2 d^2}{\gamma^4 (1-2p)^2},$$

then the *accuracy of retraining is **greater** than the accuracy of initial training*.

Some comments regarding Remark 4.10 are in order.

**Range of $n$ in Remark 4.10.** We believe that the lower bound on $n$ is tight w.r.t. $d$ and $p$, ignoring log factors. We claim this because $\Omega\left( \frac{d}{(1-2p)^2} \right)$ samples are necessary for $\mathrm{acc}(\widehat{\theta}_0) > 1 - p$ as per Theorem 4.6, and we can only expect the retraining classifier $\widehat{\theta}_1$ to be better than the initial training classifier $\widehat{\theta}_0$ if the accuracy of $\widehat{\theta}_0$'s predicted labels (which is used to train $\widehat{\theta}_1$) is more than the accuracy of the given labels $= 1 - p$. In contrast, the upper bound on $n$ may be an artifact of our analysis. Having said that, the range of $\frac{d^2}{(1-2p)^2} \lesssim n$ is not very interesting because it is far more than the number of learnable parameters $= d$.

**Effect of degree of separation $\gamma$.** As $\gamma$ increases, the minimum value of $n$ in Remark 4.10 for which we obtain an improvement with retraining reduces. This is consistent with the high-level insight of Fig. 1, viz., retraining is more beneficial when the separation between the classes is large.

## 5  Improving Label DP Training with Retraining (RT)

Motivated by our theoretical results in Section 4 which show that retraining (abbreviated as RT henceforth) can improve accuracy in the presence of label noise, we propose to apply our proposals in Section 1, viz., full RT and more importantly, *consensus-based RT* to improve local label DP training because it involves training with noisy labels. Note that this can be done **on top of any label DP mechanism** and that too **at no additional privacy cost** (both the predicted labels and originally provided noisy labels are private). Here we empirically evaluate full and consensus-based RT on four classification datasets (available on TensorFlow) trained with label DP. These include three vision datasets, namely CIFAR-10, CIFAR-100, and DomainNet (Peng et al., 2019), and one language dataset, namely AG News Subset (Zhang et al., 2015). Unless otherwise mentioned, all our empirical results here are with the standard cross-entropy loss. All the results are averaged over three runs. We only provide important experimental details here; the other details can be found in Appendix H.

**CIFAR-10/100.** We train a ResNet-18 model (from scratch) on CIFAR-10 and CIFAR-100 with label DP. Label DP training is done with the prior-based method of Ghazi et al.

(2021) – specifically, Algorithm 3 with two stages. Our training set consists of 45k examples and we assume access to a validation set with clean labels consisting of 5k examples which we use for deciding when to stop training, setting hyper-parameters, etc.[7] For CIFAR-10 and CIFAR-100 with three different values of $\epsilon$ (DP parameter; see Def. 3.1), we list the test accuracies of the baseline (i.e., the method of Ghazi et al. (2021)), full RT and consensus-based RT in Tables 1 and 2, respectively. Notice that **consensus-based RT is the clear winner**. Also, for the three values of $\epsilon$ in Table 1 (CIFAR-10), the size of the consensus set (used in consensus-based RT) is $\sim 31\%$, $55\%$ and $76\%$, respectively, of the entire training set. The corresponding numbers for Table 2 (CIFAR-100) are $\sim 11\%$, $34\%$ and $56\%$, respectively. So for small $\epsilon$ (high label noise), *consensus-based RT comprehensively outperforms full RT and the baseline with a small fraction of the training set*. Further, in Table 3, we list the accuracies of the predicted labels and the given labels over the entire (training) dataset and the accuracy of the predicted labels (which are the same as the given labels) over the consensus set for CIFAR-10 and CIFAR-100. To summarize, the accuracy of predicted labels over the consensus set is significantly more than the accuracy of predicted and given labels over the entire dataset. This gives us an idea of why consensus-based RT is much better than full RT and baseline, even though the consensus set is smaller than full dataset.

*Table 1.* **CIFAR-10.** Test set accuracies (mean ± standard deviation). *Consensus-based RT is better than full RT which is better than the baseline* (initial training).

| $\epsilon$ | Baseline | Full RT | Consensus-based RT |
|---|---|---|---|
| 1 | $57.78 \pm 1.13$ | $60.07 \pm 0.63$ | $\mathbf{63.84} \pm 0.56$ |
| 2 | $79.06 \pm 0.59$ | $81.34 \pm 0.40$ | $\mathbf{83.31} \pm 0.28$ |
| 3 | $85.18 \pm 0.50$ | $86.67 \pm 0.28$ | $\mathbf{87.67} \pm 0.28$ |

*Table 2.* **CIFAR-100.** Test set accuracies (mean ± standard deviation). Overall, *consensus-based RT is **significantly** better than full RT which is somewhat better than the baseline* (initial training).

| $\epsilon$ | Baseline | Full RT | Consensus-based RT |
|---|---|---|---|
| 3 | $23.53 \pm 1.01$ | $24.42 \pm 1.22$ | $\mathbf{29.98} \pm 1.11$ |
| 4 | $44.53 \pm 0.81$ | $46.99 \pm 0.66$ | $\mathbf{51.30} \pm 0.98$ |
| 5 | $55.75 \pm 0.36$ | $56.98 \pm 0.43$ | $\mathbf{59.47} \pm 0.26$ |

One may wonder how an increase in the number of model parameters affects results because of potential overfitting to label noise. So next, we train ResNet-34 (which has

---

[7]In practice, we do not need access to the validation set. Instead, it can be stored by a secure agent which returns us a private version of the validation accuracy and this will not be too far off from the true validation accuracy when the validation set is large enough. We also show some results without a validation set in Appendix K.

*Table 3.* **CIFAR-10 (top) and CIFAR-100 (bottom).** Accuracies of predicted labels and given labels over the entire (training) dataset and accuracies of predicted labels over the consensus set. Note that the *accuracy over the consensus set ≫ accuracy of over the entire dataset* (with both predicted and given labels). This gives us an idea of why consensus-based RT is much better than full RT and baseline, even though the consensus set is smaller than the full dataset (∼ 31%, 55% and 76% of the full dataset for $\epsilon = 1, 2$ and $3$ in the case of CIFAR-10, and ∼ 11%, 34% and 56% of the full dataset for $\epsilon = 3, 4$ and $5$ in the case of CIFAR-100).

| | $\epsilon$ | Acc. of *predicted* labels on *full* dataset | Acc. of *given* labels on *full* dataset | Acc. of *predicted* labels on *consensus* set |
|---|---|---|---|---|
| CIFAR-10 | 1 | $59.30 \pm 0.74$ | $32.61 \pm 0.74$ | **76.17** $\pm 0.15$ |
| | 2 | $81.62 \pm 0.18$ | $57.11 \pm 0.05$ | **92.65** $\pm 0.22$ |
| | 3 | $89.28 \pm 0.35$ | $76.73 \pm 0.12$ | **95.94** $\pm 0.23$ |

| | $\epsilon$ | Acc. of *predicted* labels on *full* dataset | Acc. of *given* labels on *full* dataset | Acc. of *predicted* labels on *consensus* set |
|---|---|---|---|---|
| CIFAR-100 | 3 | $24.90 \pm 0.92$ | $22.35 \pm 0.41$ | **76.09** $\pm 0.85$ |
| | 4 | $50.85 \pm 0.82$ | $46.32 \pm 0.34$ | **91.59** $\pm 1.24$ |
| | 5 | $66.51 \pm 0.02$ | $68.09 \pm 0.33$ | **94.83** $\pm 0.15$ |

*Table 4.* **CIFAR-100 w/ ResNet-34.** Test set accuracies (mean ± standard deviation). Just like Table 2 (ResNet-18), consensus-based RT is the clear winner here. But the performance in case of ResNet-34 is worse than ResNet-18 due to more overfitting to noise because of more parameters; this is expected.

| $\epsilon$ | Baseline | Full RT | Consensus-based RT |
|---|---|---|---|
| 4 | $37.53 \pm 1.58$ | $39.33 \pm 1.37$ | **43.87** $\pm 1.62$ |
| 5 | $51.13 \pm 0.69$ | $52.33 \pm 0.38$ | **55.43** $\pm 0.42$ |

nearly six times the number of parameters of ResNet-18) on CIFAR-100; the setup is exactly the same as our previous ResNet-18 experiments. Additionally, one may wonder if retraining is useful when a label noise-robust technique is used during initial training, i.e., on top of the method of Ghazi et al. (2021). To that end, we train ResNet-34 on CIFAR-100 with (*i*) the popular noise-correcting technique of *forward correction* (Patrini et al., 2017) applied to the first stage of Ghazi et al. (2021) (it is not clear how to apply it to the second stage), and (*ii*) the noise-robust *symmetric cross-entropy (CE) loss function* of Wang et al. (2019) with $\alpha = 0.8, \beta = 0.2$ used instead of the standard CE loss function. We show the results for ResNet-34 without any noise-robust method, forward correction used initially, and with the symmetric CE loss function in Tables 4, 5, and 6, respectively, for $\epsilon = \{4, 5\}$; please see the detailed discussion in the captions. In summary, consensus-based RT is pretty effective even with overfitting by ResNet-34. Also, more importantly, **consensus-based RT is beneficial even after noise-robust initial training**.

**DomainNet** (https://www.tensorflow.org/datasets/catalog/domainnet). This is an image classification dataset much larger than CIFAR where the images belong to one of 345 classes. Here we do linear probing (i.e., fitting a softmax layer) on top of a ResNet-50 pretrained on ImageNet. We reserve 10% of the entire training set for validation and use the rest for training with label DP. Just like the CIFAR experiments, we assume

*Table 5.* **CIFAR-100 w/ ResNet-34: noise-robust *forward correction* (Patrini et al., 2017) used initially** (in the baseline). Test set accuracies (mean ± standard deviation). We see that *retraining (especially, consensus-based RT) yields improvement even after noise-robust initial training*, although the amount of improvement is less compared to Table 4 (no noise-correction).

| $\epsilon$ | Baseline | Full RT | Consensus-based RT |
|---|---|---|---|
| 4 | $41.80 \pm 0.90$ | $42.63 \pm 0.58$ | **46.53** $\pm 0.76$ |
| 5 | $53.83 \pm 0.83$ | $54.43 \pm 0.48$ | **56.60** $\pm 0.43$ |

*Table 6.* **CIFAR-100 w/ ResNet-34: noise-robust *symmetric CE loss* (Wang et al., 2019) used instead of standard CE loss.** Test set accuracies (mean ± standard deviation). We see that *retraining (especially, consensus-based RT) yields improvement even with a noise-robust loss function*.

| $\epsilon$ | Baseline | Full RT | Consensus-based RT |
|---|---|---|---|
| 4 | $37.07 \pm 2.03$ | $38.17 \pm 2.03$ | **43.20** $\pm 1.77$ |
| 5 | $53.10 \pm 0.54$ | $53.40 \pm 0.33$ | **56.13** $\pm 0.25$ |

that the validation set comes with clean labels. In Table 7, we show results for $\epsilon = \{3, 4\}$ when label DP training is done with the method of Ghazi et al. (2021) with two stages (same as the CIFAR experiments). Observe that **consensus-based RT is the clear winner**. Further, in Tables 8 and 9, we show the corresponding results when the noise-robust techniques of *forward* and *backward* correction (Patrini et al., 2017) are applied to the first stage of Ghazi et al. (2021) in initial training (it is not clear how to apply forward and backward correction to the second stage). As expected, forward & backward correction lead to better initial model performance (compared to no correction). The main thing to note however is that **consensus-based RT yields significant gains even after noise-robust initial training**, consistent with our earlier results.[8]

---

[8]It is worth noting that for $\epsilon = 3$, consensus-based RT leads to similar accuracy with and without noise correction.

*Table 7.* **DomainNet.** Test set accuracies (mean ± standard deviation). *Consensus-based RT leads to very significant improvement.*

| $\epsilon$ | Baseline | Full RT | Consensus-based RT |
|---|---|---|---|
| 3 | $23.60 \pm 0.92$ | $29.23 \pm 1.03$ | $\mathbf{36.30} \pm 0.75$ |
| 4 | $48.25 \pm 0.05$ | $52.10 \pm 0.10$ | $\mathbf{57.40} \pm 0.20$ |

*Table 8.* **DomainNet: noise-robust *forward correction* used initially.** Test set accuracies (mean ± standard deviation). *Consensus-based RT yields significant improvement even after noise-robust initial training.*

| $\epsilon$ | Baseline | Full RT | Consensus-based RT |
|---|---|---|---|
| 3 | $31.23 \pm 0.56$ | $33.30 \pm 0.65$ | $\mathbf{36.07} \pm 0.78$ |
| 4 | $58.50 \pm 0.08$ | $58.63 \pm 0.12$ | $\mathbf{61.80} \pm 0.08$ |

*Table 9.* **DomainNet: noise-robust *backward correction* used initially.** Test set accuracies (mean ± standard deviation). Once again, *consensus-based RT yields significant improvement even after noise-robust initial training.*

| $\epsilon$ | Baseline | Full RT | Consensus-based RT |
|---|---|---|---|
| 3 | $30.17 \pm 0.61$ | $31.47 \pm 0.74$ | $\mathbf{35.03} \pm 0.78$ |
| 4 | $56.63 \pm 0.37$ | $56.80 \pm 0.37$ | $\mathbf{60.47} \pm 0.46$ |

Finally, we show results on a language dataset.

**AG News Subset** (`https://www.tensorflow.org/datasets/catalog/ag_news_subset`). This is a news article classification dataset consisting of 4 categories – world, sports, business or sci/tech. Just like the previous experiments, we keep 10% of the full training set for validation which we assume comes with clean labels, and use the rest for training with label DP. We use the small BERT model available in TensorFlow and the BERT English uncased preprocessor; links to both of these are in Appendix H. We pool the output of the BERT encoder, add a dropout layer with probability = 0.2, followed by a softmax layer. We fine-tune the full model. Here label DP training is done with randomized response. We list the test accuracies of the baseline (i.e., randomized response), full RT and consensus-based RT in Table 10 for three different values of $\epsilon$. Once again, **consensus-based RT is the clear winner**. For the three values of $\epsilon$ in Table 10, the size of the consensus set is ~ 28%, 32% and 38%, respectively, of the entire training set. So here, *consensus-based RT appreciably outperforms full RT and baseline with less than two-fifths of the entire training set*. In Table 11 (Appendix I), we list the accuracies of the predicted & given labels over the full dataset and the consensus set; the observations are similar to Table 3 giving us an idea of why consensus-based RT performs the best.

So in summary, *consensus-based retraining is a* **straightforward post-processing step** *that can be applied on top of a*

*Table 10.* **AG News Subset.** Test set accuracies (mean ± standard deviation). *Consensus-based RT is better than full RT which is better than the baseline.*

| $\epsilon$ | Baseline | Full RT | Consensus-based RT |
|---|---|---|---|
| 0.3 | $54.54 \pm 0.97$ | $60.03 \pm 2.90$ | $\mathbf{65.91} \pm 1.93$ |
| 0.5 | $69.21 \pm 0.31$ | $75.63 \pm 1.08$ | $\mathbf{80.95} \pm 1.47$ |
| 0.8 | $79.10 \pm 1.43$ | $82.19 \pm 1.54$ | $\mathbf{84.26} \pm 1.03$ |

*base method to* **significantly improve** *its performance in the presence of noisy labels*.

**Additional empirical results in the Appendix.** In Appendix J, we show that consensus-based RT outperforms retraining on samples for which the model is the most confident; this is similar to self-training's method of sample selection in the semi-supervised setting. In Appendix K, we show that RT is beneficial even without a validation set. Finally, going beyond label DP, we show that consensus-based RT is beneficial in the presence of human annotation errors which can be thought of as "real" label noise in Appendix L.

## 6   Conclusion

In this work, we provided the first theoretical result showing retraining with predicted hard labels can provably increase model accuracy in the presence of label noise. We also showed the efficacy of consensus-based retraining (i.e., retraining on only those samples for which the predicted label matches the given label) in improving local label DP training. We will conclude by discussing some limitations of our work which pave the way for future directions of work. Our theoretical results in this work focused on full retraining. Because consensus-based retraining worked very well empirically, we would like to analyze it theoretically in the future. Also, our theoretical results are under the uniform label noise model. In the future, we would like to analyze retraining under non-uniform label noise models. We also hope to test our ideas on larger-scale models and datasets.

## Acknowledgments

Adel Javanmard is supported in part by the NSF Award DMS-2311024, the Sloan fellowship in Mathematics, an Adobe Faculty Research Award, an Amazon Faculty Research Award, and an iORB grant from USC Marshall School of Business. Sujay Sanghavi is partially supported by NSF EnCORE Tripods (2217069) and NSF AI Institute for the Foundations of Machine Learning (2019844). The authors are grateful to anonymous reviewers for their feedback on improving this paper.

## Impact Statement

This paper presents work whose goal is to advance the field of machine learning. There are potential societal consequences of our work, none of which we feel must be specifically highlighted here.

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

# Appendix

**Table of Contents**

# A Problem Setting of Figure 1

The setting is exactly the same as the problem setting in Section 4 with $d = 50$, $\boldsymbol{\mu} = \gamma \boldsymbol{e}_1$, $\boldsymbol{\Sigma} = \boldsymbol{I} - \boldsymbol{e}_1 \boldsymbol{e}_1^\mathsf{T}$, and $p = 0.4$. In the direction of $\boldsymbol{\mu}$, we have $\langle \boldsymbol{x}, \boldsymbol{\mu} \rangle = y(1+u) \|\boldsymbol{\mu}\|_{\ell_2}^2 = y(1+u)\gamma^2$. We consider balanced classes (i.e., $\pi_+ = \pi_- = \frac{1}{2}$) and choose $u \sim \mathrm{Unif}[0, 4]$, the uniform distribution over the interval $[0, 4]$. In Figure 1a, $\gamma^2 = 0.5$ (large separation) and in Figure 1b, $\gamma^2 = 0.3$ (small separation). The number of training samples in each case is 300 and the retraining is done on the same training set on which the model is initially trained. The learned classifiers from initial training and retraining are the same as in Section 4 (i.e., Equations (4.3) and (4.8), respectively). Finally, the test accuracy of both initial training and retraining in Figure 1b is 68%.

# B Proof of Theorem 4.1

We first prove the lower bound $\alpha_0(\boldsymbol{x})$ and then the upper bound $\widetilde{\alpha}_0(\boldsymbol{x})$.

## B.1 Lower Bound $\alpha_0(\mathbf{x})$

Let $\xi_i = y_i \widehat{y}_i$. So $\xi_i = 1$ with probability $1 - p$ and $\xi_i = -1$ with probability $p$. Under our data model, for each sample we can write $\boldsymbol{x}_i = y_i(1 + u_i)\boldsymbol{\mu} + \boldsymbol{\Sigma}^{1/2} \boldsymbol{z}_i$ with $\boldsymbol{z}_i \sim \mathcal{N}(0, \boldsymbol{I}_d)$. Hence,

$$n\widehat{\boldsymbol{\theta}}_0 = \sum_{i \in [n]} \widehat{y}_i \boldsymbol{x}_i = \Big( \sum_{i \in [n]} \xi_i(1 + u_i) \Big)\boldsymbol{\mu} + \sum_{i \in [n]} \widehat{y}_i \boldsymbol{\Sigma}^{1/2} \boldsymbol{z}_i \,.$$

We also note that

$$\mathbb{P}(\mathrm{sign}(\langle \boldsymbol{x}, \widehat{\boldsymbol{\theta}}_0 \rangle) \ne y) = \mathbb{P}(y\langle \boldsymbol{x}, \widehat{\boldsymbol{\theta}}_0 \rangle \le 0) = \mathbb{P}\Bigg( \Big( \sum_{i \in [n]} \xi_i(1 + u_i) \Big)\langle y\boldsymbol{x}, \boldsymbol{\mu} \rangle + \langle \boldsymbol{x}, \sum_{i \in [n]} y\widehat{y}_i \boldsymbol{\Sigma}^{1/2} \boldsymbol{z}_i \rangle \le 0 \Bigg)$$

Define $\tilde{\boldsymbol{z}} := \sum_{i \in [n]} y\widehat{y}_i \boldsymbol{\Sigma}^{1/2} \boldsymbol{z}_i$. Conditioning on $\{\widehat{y}_i\}_{i \in [n]}$ we have $\tilde{\boldsymbol{z}} \sim \mathcal{N}(\boldsymbol{0}, n\boldsymbol{\Sigma})$, and so $\langle \boldsymbol{x}, \tilde{\boldsymbol{z}} \rangle \sim \mathcal{N}(\boldsymbol{0}, n \|\boldsymbol{\Sigma}^{1/2}\boldsymbol{x}\|_{\ell_2}^2)$.

We write

$$\mathbb{P}\Bigg( \Big( \sum_{i \in [n]} \xi_i(1 + u_i) \Big)\langle y\boldsymbol{x}, \boldsymbol{\mu} \rangle + \langle \boldsymbol{x}, \tilde{\boldsymbol{z}} \rangle \le 0 \ \Big| \ \{\xi_i, y_i, u_i\}_{i \in [n]} \Bigg) = \Phi\Bigg( -\frac{(\sum_{i \in [n]} \xi_i(1 + u_i))\langle y\boldsymbol{x}, \boldsymbol{\mu} \rangle}{\sqrt{n} \|\boldsymbol{\Sigma}^{1/2}\boldsymbol{x}\|_{\ell_2}} \Bigg),$$

where $\Phi(t) = \frac{1}{\sqrt{2\pi}} \int_{-\infty}^{t} e^{-s^2/2} \mathrm{d}s$ denotes the cdf of standard normal distribution.

Combining the previous two equations, we arrive at

$$\begin{aligned}
\mathbb{P}(\mathrm{sign}(\langle \boldsymbol{x}, \widehat{\boldsymbol{\theta}}_0 \rangle) \ne y) &= \mathbb{E}\Bigg[ \Phi\Bigg( -\frac{(\sum_{i \in [n]} \xi_i(1 + u_i))\langle y\boldsymbol{x}, \boldsymbol{\mu} \rangle}{\sqrt{n} \|\boldsymbol{\Sigma}^{1/2}\boldsymbol{x}\|_{\ell_2}} \Bigg) \Bigg] \\
&\ge \mathbb{E}\Bigg[ \Phi\Bigg( -\frac{|\sum_{i \in [n]} \xi_i(1 + u_i)| \, |\langle \boldsymbol{x}, \boldsymbol{\mu} \rangle|}{\sqrt{n} \|\boldsymbol{\Sigma}^{1/2}\boldsymbol{x}\|_{\ell_2}} \Bigg) \Bigg] \\
&\ge \Phi\Bigg( -\frac{\mathbb{E}[|\sum_{i \in [n]} \xi_i(1 + u_i)|] \, |\langle \boldsymbol{x}, \boldsymbol{\mu} \rangle|}{\sqrt{n} \|\boldsymbol{\Sigma}^{1/2}\boldsymbol{x}\|_{\ell_2}} \Bigg)
\end{aligned} \tag{B.1}$$

where the expectation is with respect to $\{\xi_i, u_i\}_{i \in [n]}$ and the last step follows by Jensen's inequality and the fact that $\Phi(t)$ is a convex function for $t \le 0$.

We next note that by Cauchy–Schwarz inequality,

$$
\mathbb{E}\left[\left|\sum_i \xi_i(1+u_i)\right|\right] \le \sqrt{\mathbb{E}[(\sum_i \xi_i(1+u_i))^2]}
$$

$$
= \sqrt{\sum_{i,j} \mathbb{E}[\xi_i \xi_j (1+u_i)(1+u_j)]}
$$

$$
= \sqrt{\sum_i \mathbb{E}[(1+u_i)^2] + \sum_{i\ne j} \mathbb{E}[\xi_i]\,\mathbb{E}[\xi_j]\,\mathbb{E}[1+u_i]\,\mathbb{E}[1+u_j]}
$$

$$
= \sqrt{5n + 4n(n-1)(1-2p)^2} \le \sqrt{5n} + \sqrt{5}n(1-2p),
$$

where we used that $\xi_i, \xi_j$ are independent for $i \ne j$, and independent from $u_i$'s. In addition, $\mathbb{E}[\xi_i] = \mathbb{E}[y_i\widehat{y_i}] = \mathbb{P}(y_i = \widehat{y_i}) - \mathbb{P}(y_i \ne \widehat{y_i}) = 1 - 2p$. Also $\mathbb{E}[u_i] \le 1$ and $\mathbb{E}[u_i^2] \le 2$, since $u_i$ has unit sub-gaussian norm. Hence, we get

$$
\mathbb{P}(\mathrm{sign}(\langle \boldsymbol{x}, \widehat{\boldsymbol{\theta}}_0\rangle) \ne y) \ge \Phi\left(-\frac{\sqrt{5}(1+\sqrt{n}(1-2p))\,|\langle \boldsymbol{x}, \boldsymbol{\mu}\rangle|}{\left\|\boldsymbol{\Sigma}^{1/2}\boldsymbol{x}\right\|_{\ell_2}}\right). \tag{B.2}
$$

We next use a classical lower bound on $\Phi$. Let $\Phi^c(t) = 1 - \Phi(t) = \Phi(-t)$. Using Equation (7.1.13) of Abramowitz & Stegun (1968), we have for any $t > 0$:

$$
\Phi^c(t) > \sqrt{\frac{2}{\pi}}\left(\frac{e^{-\frac{t^2}{2}}}{t + \sqrt{t^2+4}}\right).
$$

Since $t + \sqrt{t^2+4} < 2(t+1)$ for $t > 0$, we get:

$$
\Phi^c(t) > \frac{1}{\sqrt{2\pi}}\left(\frac{e^{-\frac{t^2}{2}}}{t+1}\right) > \frac{1}{2\sqrt{2\pi}}\exp(-t^2), \tag{B.3}
$$

where we used that $t + 1 \le t^2 + 2 \le 2e^{t^2/2}$. Combining (B.3) and (B.2) we get

$$
\mathbb{P}(\mathrm{sign}(\langle \boldsymbol{x}, \widehat{\boldsymbol{\theta}}_0\rangle) \ne y) \ge \frac{1}{2\sqrt{2\pi}}\exp\left(-\frac{5(1+\sqrt{n}(1-2p))^2\langle \boldsymbol{x}, \boldsymbol{\mu}\rangle^2}{\left\|\boldsymbol{\Sigma}^{1/2}\boldsymbol{x}\right\|_{\ell_2}^2}\right),
$$

which completes the derivation of $\alpha_0(\boldsymbol{x})$.

## B.2 Upper Bound $\widetilde{\alpha}_0(\mathbf{x})$

We continue from (B.1), which reads

$$
\mathbb{P}(\mathrm{sign}(\langle \boldsymbol{x}, \widehat{\boldsymbol{\theta}}_0\rangle) \ne y) = \mathbb{E}\left[\Phi\left(-\frac{(\sum_{i\in[n]}\xi_i(1+u_i))\langle y\boldsymbol{x}, \boldsymbol{\mu}\rangle}{\sqrt{n}\left\|\boldsymbol{\Sigma}^{1/2}\boldsymbol{x}\right\|_{\ell_2}}\right)\right], \tag{B.4}
$$

where the expectation is with respect to the randomness in training data, namely $u_i$ and $\xi_i$ for $i \in [n]$, while the test point $(\boldsymbol{x}, y)$ is fixed.

Note that under our data model, $\langle y\boldsymbol{x}, \boldsymbol{\mu}\rangle = (1+u)\|\boldsymbol{\mu}\|_{\ell_2}^2 = (1+u)\gamma^2 > 0$ since $u$ is a non-negative random variable. We next define the following event:

$$
\mathcal{E}_0 := \left\{\sum_{i\in[n]}\xi_i(1+u_i) \ge n(1-2p)/2\right\}. \tag{B.5}
$$

Recall that $\mathbb{E}[\xi_i] = 1 - 2p$ and by the Hoeffding-type inequality for sub-gaussian random variables,

$$
\mathbb{P}\left(\left|\sum_i \xi_i(1+u_i) - n(1-2p)(1+\mathbb{E}[u_i])\right| \ge t\right) \le 2\exp\left(-\frac{t^2}{8n}\right),
$$

(Note that $\|\xi_i(1 + u_i)\|_{\psi_2} \le 1 + \|u_i\|_{\psi_2} = 2$.) By choosing $t = n(1 - 2p)/2$, and using the fact that $u_i > 0$ for all $i$ we get

$$\mathbb{P}\left(\sum_i \xi_i(1 + u_i) < n(1 - 2p)/2\right) \le 2\exp\left(-\frac{n}{32}(1 - 2p)^2\right).$$

Hence, $\mathbb{P}(\mathcal{E}_0^c) \le 2\exp\left(-\frac{n}{32}(1 - 2p)^2\right)$.

We then have

$$
\begin{aligned}
\mathbb{E}\left[\Phi\left(-\frac{(\sum_{i\in[n]}\xi_i(1 + u_i))\langle y\boldsymbol{x}, \boldsymbol{\mu}\rangle}{\sqrt{n}\left\|\boldsymbol{\Sigma}^{1/2}\boldsymbol{x}\right\|_{\ell_2}}\right)\right] &= \mathbb{E}\left[\Phi\left(-\frac{(\sum_{i\in[n]}\xi_i(1 + u_i))\langle y\boldsymbol{x}, \boldsymbol{\mu}\rangle}{\sqrt{n}\left\|\boldsymbol{\Sigma}^{1/2}\boldsymbol{x}\right\|_{\ell_2}}\right)(\mathbb{1}_{\mathcal{E}_0} + \mathbb{1}_{\mathcal{E}_0^c})\right] \\
&\le \mathbb{E}\left[\Phi\left(-\frac{n(1 - 2p)\langle y\boldsymbol{x}, \boldsymbol{\mu}\rangle}{2\sqrt{n}\left\|\boldsymbol{\Sigma}^{1/2}\boldsymbol{x}\right\|_{\ell_2}}\right)\mathbb{1}_{\mathcal{E}_0}\right] + \mathbb{P}(\mathcal{E}_0^c) \\
&= \Phi\left(-\frac{n(1 - 2p)\langle y\boldsymbol{x}, \boldsymbol{\mu}\rangle}{2\sqrt{n}\left\|\boldsymbol{\Sigma}^{1/2}\boldsymbol{x}\right\|_{\ell_2}}\right)\mathbb{P}(\mathcal{E}_0) + \mathbb{P}(\mathcal{E}_0^c) \\
&\le \Phi\left(-\frac{n(1 - 2p)\langle y\boldsymbol{x}, \boldsymbol{\mu}\rangle}{2\sqrt{n}\left\|\boldsymbol{\Sigma}^{1/2}\boldsymbol{x}\right\|_{\ell_2}}\right) + \mathbb{P}(\mathcal{E}_0^c),
\end{aligned}
$$

where in the first inequality we used the observation that $\langle y\boldsymbol{x}, \boldsymbol{\mu}\rangle \ge 0$ as explained above, the definition of $\mathcal{E}_0$ and that $\Phi(-z)$ is a decreasing function in $z$.

We next recall the tail bound of normal distribution in Equation (7.1.13) of Abramowitz & Stegun (1968):

$$\Phi^c(t) \le \sqrt{\frac{2}{\pi}}\frac{e^{-t^2/2}}{t + \sqrt{t^2 + \frac{8}{\pi}}} \le \frac{1}{2}e^{-t^2/2},$$

for all $t \ge 0$, where $\Phi^c(t) = 1 - \Phi(t) = \Phi(-t)$ is the complementary CDF of normal variables. Using this bound and the bound we derived on $\mathbb{P}(\mathcal{E}_0^c)$ we get

$$\mathbb{E}\left[\Phi\left(-\frac{(\sum_{i\in[n]}\xi_i(1 + u_i))\langle y\boldsymbol{x}, \boldsymbol{\mu}\rangle}{\sqrt{n}\left\|\boldsymbol{\Sigma}^{1/2}\boldsymbol{x}\right\|_{\ell_2}}\right)\right] \le \frac{1}{2}\exp\left(-\frac{n(1 - 2p)^2\langle y\boldsymbol{x}, \boldsymbol{\mu}\rangle^2}{8\left\|\boldsymbol{\Sigma}^{1/2}\boldsymbol{x}\right\|_{\ell_2}^2}\right) + 2\exp\left(-\frac{n}{32}(1 - 2p)^2\right).$$

By invoking (B.4), we obtain the desired bound $\widetilde{\alpha}_0(\boldsymbol{x})$.

## C Proof of Theorem 4.2

We will first derive the lower bound on $\text{err}(\widehat{\boldsymbol{\theta}}_0)$. Using Theorem 4.1 (and because the training set $\mathcal{T}$ has measure zero), we have

$$
\begin{aligned}
\text{err}(\widehat{\boldsymbol{\theta}}_0) &\ge \mathbb{E}[\alpha_0(\boldsymbol{x})] \\
&= \frac{1}{2\sqrt{2\pi}}\mathbb{E}\left[\exp\left(-\frac{5(1 + \sqrt{n}(1 - 2p))^2\langle\boldsymbol{x}, \boldsymbol{\mu}\rangle^2}{\left\|\boldsymbol{\Sigma}^{1/2}\boldsymbol{x}\right\|_{\ell_2}^2}\right)\right] \\
&\ge \frac{1}{2\sqrt{2\pi}}\mathbb{E}\left[\exp\left(-\frac{5(1 + \sqrt{n}(1 - 2p))^2\langle\boldsymbol{x}, \boldsymbol{\mu}\rangle^2}{\left\|\boldsymbol{\Sigma}^{1/2}\boldsymbol{x}\right\|_{\ell_2}^2}\right)\Bigg| \frac{\langle\boldsymbol{x}, \boldsymbol{\mu}\rangle^2}{\left\|\boldsymbol{\Sigma}^{1/2}\boldsymbol{x}\right\|_{\ell_2}^2} \le v^2\right]\mathbb{P}\left(\frac{\langle\boldsymbol{x}, \boldsymbol{\mu}\rangle^2}{\left\|\boldsymbol{\Sigma}^{1/2}\boldsymbol{x}\right\|_{\ell_2}^2} \le v^2\right) \\
&\ge \frac{1}{2\sqrt{2\pi}}\exp\left(-5(1 + \sqrt{n}(1 - 2p))^2 v^2\right)\mathbb{P}\left(\frac{\langle\boldsymbol{x}, \boldsymbol{\mu}\rangle^2}{\left\|\boldsymbol{\Sigma}^{1/2}\boldsymbol{x}\right\|_{\ell_2}^2} \le v^2\right),
\end{aligned}
\tag{C.1}
$$

for any value of $v > 0$. We next lower bound the probability on the right-hand side. We have

$$\mathbb{P}\left(\frac{|\langle \boldsymbol{x}, \boldsymbol{\mu}\rangle|}{\left\|\boldsymbol{\Sigma}^{1/2}\boldsymbol{x}\right\|_{\ell_2}} \leq v\right) = \mathbb{P}\left(\frac{|y(1+u)\gamma^2|}{\|\boldsymbol{\Sigma}\boldsymbol{z}\|_{\ell_2}} \leq v\right)$$

$$\geq \mathbb{P}\left(\frac{(1+u)\gamma^2}{\lambda_{\min}\left\|\mathcal{P}_{\boldsymbol{\mu}}^{\perp}\boldsymbol{z}\right\|_{\ell_2}} \leq v\right)$$

$$= \mathbb{P}\left(\frac{(1+u)\gamma^2}{\left\|\mathcal{P}_{\boldsymbol{\mu}}^{\perp}\boldsymbol{z}\right\|_{\ell_2}} \leq v\lambda_{\min}\right), \tag{C.2}$$

where the first step holds since under our data model we have $\langle \boldsymbol{x}, \boldsymbol{\mu}\rangle = y(1+u)\|\boldsymbol{\mu}\|_{\ell_2}^2$ and $\boldsymbol{\Sigma}^{1/2}\boldsymbol{x} = y(1+u)\boldsymbol{\Sigma}^{1/2}\boldsymbol{\mu} + \boldsymbol{\Sigma}\boldsymbol{z} = \boldsymbol{\Sigma}\boldsymbol{z}$, given that $\boldsymbol{\Sigma}^{1/2}\boldsymbol{\mu} = 0$. Here $\mathcal{P}_{\boldsymbol{\mu}}^{\perp}$ is the projection onto the orthogonal space of $\boldsymbol{\mu}$.

Continuing from (C.2) we write

$$\mathbb{P}\left(\frac{(1+u)\gamma^2}{\left\|\mathcal{P}_{\boldsymbol{\mu}}^{\perp}\boldsymbol{z}\right\|_{\ell_2}} \leq v\lambda_{\min}\right) \geq \mathbb{P}\left((1+u)\gamma^2 \leq v\lambda_{\min}\sqrt{d/2},\ \left\|\mathcal{P}_{\boldsymbol{\mu}}^{\perp}\boldsymbol{z}\right\|_{\ell_2} \geq \sqrt{d/2}\right)$$

$$= \mathbb{P}\left((1+u)\gamma^2 \leq v\lambda_{\min}\sqrt{d/2}\right)\mathbb{P}\left(\left\|\mathcal{P}_{\boldsymbol{\mu}}^{\perp}\boldsymbol{z}\right\|_{\ell_2} \geq \sqrt{d/2}\right), \tag{C.3}$$

given that $\boldsymbol{z}$ and $u$ are independent. Next note that $\mathcal{P}_{\boldsymbol{\mu}}^{\perp}\boldsymbol{z}$ is distributed as a Gaussian vector in a $(d-1)$-dimensional space, and therefore $\left\|\mathcal{P}_{\boldsymbol{\mu}}^{\perp}\boldsymbol{z}\right\|_{\ell_2}^2$ follows the chi-squared distribution with $(d-1)$ degrees of freedom. Using the tail bound of the chi-squared distribution (see Theorem 4 of Ghosh (2021)), we get:

$$\mathbb{P}\left(\left\|\mathcal{P}_{\boldsymbol{\mu}}^{\perp}\boldsymbol{z}\right\|_{\ell_2}^2 < d/2\right) \leq e^{-d/16} \implies \mathbb{P}\left(\left\|\mathcal{P}_{\boldsymbol{\mu}}^{\perp}\boldsymbol{z}\right\|_{\ell_2} \geq \sqrt{d/2}\right) \geq 1 - e^{-d/16}. \tag{C.4}$$

We next lower bound the other term on the right-hand side of (C.3). By Markov's inequality for any non-negative random variable $X$, we have $\mathbb{P}(X \leq 2\mathbb{E}[X]) \geq 1/2$. Also $\mathbb{E}[(1+u)\gamma^2] \leq 2\gamma^2$, since $\mathbb{E}[u] \leq 1$ and therefore, by choosing $v = \frac{4\sqrt{2}\gamma^2}{\lambda_{\min}\sqrt{d}}$, we get

$$\mathbb{P}\left((1+u)\gamma^2 \leq v\lambda_{\min}\sqrt{d/2}\right) \geq 1/2. \tag{C.5}$$

Combining (C.4), (C.5), (C.3) and (C.2) with (C.1) and plugging in our choice of $v = \frac{4\sqrt{2}\gamma^2}{\lambda_{\min}\sqrt{d}}$, we arrive at

$$\mathsf{err}(\widehat{\boldsymbol{\theta}}_0) \geq \frac{1}{4\sqrt{2\pi}}\exp\left(-160(1+\sqrt{n}(1-2p))^2\frac{\gamma^4}{\lambda_{\min}^2 d}\right)\left(1 - e^{-d/16}\right).$$

Let us now derive the upper bound on $\mathsf{err}(\widehat{\boldsymbol{\theta}}_0)$. Using Theorem 4.1 (and because the training set $\mathcal{T}$ has measure zero), we have

$$\mathsf{err}(\widehat{\boldsymbol{\theta}}_0) \leq \mathbb{E}[\widetilde{\alpha}_0(\boldsymbol{x})]$$

$$= \frac{1}{2}\mathbb{E}\left[\exp\left(-\frac{n(1-2p)^2\langle \boldsymbol{x}, \boldsymbol{\mu}\rangle^2}{8\left\|\boldsymbol{\Sigma}^{1/2}\boldsymbol{x}\right\|_{\ell_2}^2}\right)\right] + 2\exp\left(-\frac{n}{32}(1-2p)^2\right). \tag{C.6}$$

Under our data model, $\boldsymbol{x} = y(1+u)\boldsymbol{\mu} + \boldsymbol{\Sigma}^{1/2}\boldsymbol{z}$. Consider the probabilistic event $\mathcal{E}_2 := \{\boldsymbol{z} : \|\boldsymbol{z}\|_{\ell_2}^2 \leq 2d\}$. For every $\eta > 0$, we have

$$\mathbb{E}\left[\exp\left(-\frac{\eta\langle \boldsymbol{x}, \boldsymbol{\mu}\rangle^2}{\left\|\boldsymbol{\Sigma}^{1/2}\boldsymbol{x}\right\|_{\ell_2}^2}\right)\right] = \mathbb{E}\left[\exp\left(-\frac{\eta\langle \boldsymbol{x}, \boldsymbol{\mu}\rangle^2}{\left\|\boldsymbol{\Sigma}^{1/2}\boldsymbol{x}\right\|_{\ell_2}^2}\right)(\mathbb{1}_{\mathcal{E}_2} + \mathbb{1}_{\mathcal{E}_2^c})\right]$$

$$\leq \exp\left(-\frac{\eta\gamma^4}{2\lambda_{\max}d}\right) + \mathbb{P}(\mathcal{E}_2^c),$$

where the inequality holds because $\langle \boldsymbol{x}, \boldsymbol{\mu} \rangle = y(1+u)\gamma^2$ due to which $|\langle \boldsymbol{x}, \boldsymbol{\mu} \rangle| \geq \gamma^2$ (given that $u$ is a non-negative random variable) and because

$$\left\| \boldsymbol{\Sigma}^{1/2} \boldsymbol{x} \right\|_{\ell_2}^2 = \|\boldsymbol{\Sigma} \boldsymbol{z}\|_{\ell_2}^2 \leq \lambda_{\max}^2 \|\boldsymbol{z}\|_{\ell_2}^2 \leq 2\lambda_{\max}^2 d,$$

when event $\mathcal{E}_2$ happens. Further, $\|\boldsymbol{z}\|_{\ell_2}^2 \sim \chi_d^2$ and so using tail bounds for $\chi_d^2$ (see Theorem 3 of Ghosh (2021)), we have $\mathbb{P}(\mathcal{E}_2^c) \leq e^{-d/8}$. Putting things together, we obtain

$$\mathbb{E}\left[ \exp\left( -\frac{\eta \langle \boldsymbol{x}, \boldsymbol{\mu} \rangle^2}{\left\| \boldsymbol{\Sigma}^{1/2} \boldsymbol{x} \right\|_{\ell_2}^2} \right) \right] \leq \exp\left( -\frac{\eta \gamma^4}{2\lambda_{\max}^2 d} \right) + e^{-d/8}.$$

Setting $\eta = n(1-2p)^2/8$ and using this in (C.6), we get

$$\mathrm{err}(\widehat{\boldsymbol{\theta}}_0) \leq \frac{1}{2} \exp\left( -\frac{n(1-2p)^2 \gamma^4}{16\lambda_{\max}^2 d} \right) + e^{-d/8} + 2\exp\left( -\frac{n}{32}(1-2p)^2 \right).$$

## D    Proof of Theorem 4.6

*Proof.* Since we are interested with dependence of sample size with respect to $d$ and $p$, we assume $\gamma := \|\boldsymbol{\mu}\|_{\ell_2} = 1$ and $\boldsymbol{\Sigma} = \boldsymbol{I} - \boldsymbol{\mu}\boldsymbol{\mu}^\top$ the projection onto the orthogonal space of $\boldsymbol{\mu}$. Also note that without loss of generality, we can assume that our estimator $\widehat{\boldsymbol{\theta}}$ has unit norm, because its norm does not affect the sign of $\langle \boldsymbol{x}, \widehat{\boldsymbol{\theta}} \rangle$. This way, $\boldsymbol{\mu}$ and $\widehat{\boldsymbol{\theta}}$ both belong to $\mathbb{S}^{d-1}$.

We first follow a standard argument to "reduce" the classification problem to a multi-way hypothesis testing problem. Let $\rho \in (0, 1)$ be an arbitrary but fixed value which we can choose. We define a $\rho$-packing of $\mathbb{S}^{d-1}$ as a set $\mathcal{M} = \{\boldsymbol{\mu}_1, \ldots, \boldsymbol{\mu}_M\} \subset \mathbb{S}^{d-1}$ such that $\langle \boldsymbol{\mu}_l, \boldsymbol{\mu}_k \rangle \leq \rho$ for $l \neq k$. Also define the $\rho$-packing number of $\mathbb{S}^{d-1}$ as

$$M(\rho, \mathbb{S}^{d-1}) := \sup\{M \in \mathbb{N} : \text{there exists a } \rho\text{-packing } \mathcal{M} \text{ of } \mathbb{S}^{d-1} \text{ with size } M\}. \tag{D.1}$$

For convenience, we define the misclassification error of a classifier $\widehat{\boldsymbol{\theta}} \in \mathbb{S}^{d-1}$ when the ground truth separator is $\boldsymbol{\mu}$ as $\mathrm{err}(\widehat{\boldsymbol{\theta}}; \boldsymbol{\mu}) = 1 - \mathrm{acc}(\widehat{\boldsymbol{\theta}}; \boldsymbol{\mu})$. As per our condition,

$$\delta \geq \sup_{\boldsymbol{\mu} \in \mathbb{S}^{d-1}} \mathrm{err}(\widehat{\boldsymbol{\theta}}; \boldsymbol{\mu}) \geq \sup_{\boldsymbol{\mu} \in \mathcal{M}} \mathrm{err}(\widehat{\boldsymbol{\theta}}; \boldsymbol{\mu}). \tag{D.2}$$

In order to further lower bound the right hand side, we let $I$ be a random variable uniformly distributed on the hypothesis set $\{1, 2, \ldots, M\}$ and consider the case of $\boldsymbol{\mu} = \boldsymbol{\mu}_I$. We also define $\widehat{I}$ as the index of the element in $\mathcal{M}$ with maximum inner product with $\widehat{\boldsymbol{\theta}}$ (it does not matter how we break ties). Under our data model we have $\langle \boldsymbol{x}, \boldsymbol{\mu}_I \rangle = y(1+u)\|\boldsymbol{\mu}_I\|_{\ell_2}^2 = y(1+u)$ and since $u$ is a non-negative random variable, $y = \mathrm{sign}(\langle \boldsymbol{x}, \boldsymbol{\mu}_I \rangle)$. We then have

$$\sup_{\boldsymbol{\mu} \in \mathcal{M}} \mathrm{err}(\widehat{\boldsymbol{\theta}}; \boldsymbol{\mu}) \geq \max_{i \in [M]} \mathbb{P}\left( \mathrm{sign}(\langle \boldsymbol{x}, \boldsymbol{\mu}_I \rangle) \neq \mathrm{sign}(\langle \boldsymbol{x}, \widehat{\boldsymbol{\theta}} \rangle) \Big| I = i \right)$$

$$\geq \frac{1}{M} \sum_{i=1}^{M} \mathbb{P}\left( \mathrm{sign}(\langle \boldsymbol{x}, \boldsymbol{\mu}_I \rangle) \neq \mathrm{sign}(\langle \boldsymbol{x}, \widehat{\boldsymbol{\theta}} \rangle) \Big| I = i \right)$$

$$\geq \Phi\left( -2\sqrt{\frac{1+\rho}{1-\rho}} \right) \frac{1}{M} \sum_{i=1}^{M} \mathbb{P}(\widehat{I} \neq i | I = i) \tag{D.3}$$

$$= \Phi\left( -2\sqrt{\frac{1+\rho}{1-\rho}} \right) \mathbb{P}(\widehat{I} \neq I), \tag{D.4}$$

with $\Phi$ denoting the CDF of a standard normal variable. Equation (D.3) above follows from the lemma below.

**Lemma D.1.** *For any $i \in [M]$, we have*

$$\mathbb{P}\left( \mathrm{sign}(\langle \boldsymbol{x}, \boldsymbol{\mu}_I \rangle) \neq \mathrm{sign}(\langle \boldsymbol{x}, \widehat{\boldsymbol{\theta}} \rangle) \Big| I = i \right) \geq \Phi\left( -2\sqrt{\frac{1+\rho}{1-\rho}} \right) \mathbb{P}(\widehat{I} \neq i | I = i).$$

Combining (D.2) and (D.4), we obtain that

$$\delta_0 := \frac{\delta}{\Phi\left(-2\sqrt{\frac{1+\rho}{1-\rho}}\right)} \geq \mathbb{P}(\widehat{I} \neq I). \tag{D.5}$$

Next recall the set $\mathcal{T} := \{(\boldsymbol{x}_j, \widehat{y}_j)\}_{j \in [n]}$, and let $\boldsymbol{X} = [\boldsymbol{x}_1^T, \ldots, \boldsymbol{x}_n^T]^T \in \mathbb{R}^{n \times d}$ and $\widehat{\boldsymbol{y}} = [\widehat{y}_1, \ldots, \widehat{y}_n]^T$. By an application of Fano's inequality, with conditioning on $\boldsymbol{X}$ (see e.g., Section 2.3 of Scarlett & Cevher (2019)) we have

$$\mathcal{I}(I; \widehat{I}|\boldsymbol{X}) \geq (1 - \delta_0) \log(M(\rho, \mathbb{S}^{d-1})) - \log 2, \tag{D.6}$$

where $\mathcal{I}(I; \widehat{I}|\boldsymbol{X})$ represents the conditional mutual information between $I$ and $\widehat{I}$. Using the fact that $I \to \widehat{\boldsymbol{y}} \to \widehat{I}$ forms a Markov chain conditioned on $\boldsymbol{X}$, and by an application of the data processing inequality we have:

$$\mathcal{I}(I; \widehat{I}|\boldsymbol{X}) \leq \mathcal{I}(I; \widehat{\boldsymbol{y}}|\boldsymbol{X}) \tag{D.7}$$

We will now upper bound $\mathcal{I}(I; \widehat{\boldsymbol{y}}|\boldsymbol{X})$. Let $w_j := \frac{1}{2}\big(\mathrm{sign}(\langle \boldsymbol{x}_j, \boldsymbol{\mu}_I \rangle) + 1\big)$ be the 0-1 version of the actual label of $\boldsymbol{x}_j$, viz., $\mathrm{sign}(\langle \boldsymbol{x}_j, \boldsymbol{\mu}_I \rangle)$. As per our setting, we have $\widehat{y}_j = 2\big(w_j \oplus z_j\big) - 1$ where $z_j \sim \mathrm{Bernoulli}(p)$ and $\oplus$ denotes modulo-2 addition. Since the noise variables $z_j$ are independent and $\widehat{y}_j$ depends on $(I, \boldsymbol{X})$ only through $w_j = \frac{1}{2}\big(\mathrm{sign}(\langle \boldsymbol{x}_j, \boldsymbol{\mu}_I \rangle) + 1\big)$, by using the tensorization property of the mutual information (see e.g., Lemma 2, part (iii) of Scarlett & Cevher (2019)), we have

$$\mathcal{I}(I; \boldsymbol{y}|\boldsymbol{X}) \leq \sum_{j=1}^{n} \mathcal{I}\big(w_j; \widehat{y}_j\big) \leq n(\log 2 - H_2(p)), \tag{D.8}$$

where the second inequality follows since $\widehat{y}_j$ is generated by passing $w_j$ through a binary symmetric channel, which has capacity $\log 2 - H_2(p)$ with $H_2(p) := -p \log p - (1-p) \log(1-p)$ denoting the binary entropy function.

We next use the lemma below to further upper bound the right-hand side of (D.8).

**Lemma D.2.** *For a discrete probability distribution, consider the entropy function given by*

$$H(p_1, \ldots, p_k) = \sum_{i=1}^{k} p_k \log(1/p_k).$$

*We have the following bound:*

$$H(p_1, \ldots, p_k) \geq \log k - k \sum_{i=1}^{k} (p_i - 1/k)^2.$$

Using Lemma D.2 with $k = 2$ we obtain $\log 2 - H_2(p) \leq 4(p - 1/2)^2 = (1 - 2p)^2$, which along with (D.8), (D.7) and (D.6) gives

$$\frac{n(1-2p)^2 + \log 2}{1 - \delta_0} \geq \log(M(\rho, \mathbb{S}^{d-1})). \tag{D.9}$$

In our next lemma, we lower bound $M(\rho, \mathbb{S}^{d-1})$.

**Lemma D.3.** *Recall the definition of $\rho$-packing number of $\mathbb{S}^{d-1}$ given by (D.1). We have the following bound:*

$$M(\rho, \mathbb{S}^{d-1}) \geq \exp\left(\frac{d\rho^2}{2}\right).$$

Using Lemma D.3 along with (D.9), we obtain the following lower bound on the sample complexity:

$$n \geq \frac{\frac{\rho^2}{2}(1 - \delta_0)d - \log 2}{(1 - 2p)^2}, \quad \text{with } \delta_0 = \frac{\delta}{\Phi\left(-2\sqrt{\frac{1+\rho}{1-\rho}}\right)}.$$

Note that $\rho \in (0, 1)$ can be set arbitrarily. Since our claim is on the order of $n$, the specific value of $\rho$ can be chosen based on $\delta$ to ensure $\frac{\rho^2}{2}(1 - \delta_0)$ is of the order of $(1 - \delta)$. This finishes the proof of Theorem 4.6. $\square$

We will now prove Lemmas D.1, D.2 and D.3.

**Proof of Lemma D.1.** Let us consider the event $\widehat{I} \neq i$, given that $I = i$. We note that if $\widehat{I} \neq I$, then $\langle \widehat{\boldsymbol{\theta}}, \boldsymbol{\mu}_I \rangle \leq \sqrt{\frac{1+\rho}{2}}$ or equivalently, the angle between $\widehat{\boldsymbol{\theta}}$ and $\boldsymbol{\mu}_I$ is $\geq b := \cos^{-1}\left(\sqrt{\frac{1+\rho}{2}}\right)$. Otherwise, by definition of $\widehat{I}$ we have $\langle \widehat{\boldsymbol{\theta}}, \boldsymbol{\mu}_{\widehat{I}} \rangle \geq \langle \widehat{\boldsymbol{\theta}}, \boldsymbol{\mu}_I \rangle > \sqrt{\frac{1+\rho}{2}}$, and therefore the angle between $\boldsymbol{\mu}_{\widehat{I}}$ and $\boldsymbol{\mu}_I$ is $< 2b$. Noting that $\cos(2b) = 2\cos^2(b) - 1 = \rho$, we would then have $\langle \boldsymbol{\mu}_{\widehat{I}}, \boldsymbol{\mu}_I \rangle > \rho$, which is a contradiction since $\mathcal{M}$ forms a $\rho$-packing. We proceed by writing

$$\mathbb{P}\left(\text{sign}(\langle \boldsymbol{x}, \boldsymbol{\mu}_I \rangle) \neq \text{sign}(\langle \boldsymbol{x}, \widehat{\boldsymbol{\theta}} \rangle) \middle| I = i\right) \geq \mathbb{P}\left(\text{sign}(\langle \boldsymbol{x}, \boldsymbol{\mu}_I \rangle) \neq \text{sign}(\langle \boldsymbol{x}, \widehat{\boldsymbol{\theta}} \rangle) \middle| I = i, \widehat{I} \neq i\right) \mathbb{P}(\widehat{I} \neq i | I = i).$$

As discussed above, on the event that $I = i, \widehat{I} \neq i$, we have $\theta := \langle \widehat{\boldsymbol{\theta}}, \boldsymbol{\mu}_i \rangle \leq \sqrt{\frac{1+\rho}{2}}$. Consider the decomposition $\boldsymbol{x} = \langle \boldsymbol{x}, \boldsymbol{\mu}_i \rangle \boldsymbol{\mu}_i + \mathcal{P}_{\boldsymbol{\mu}_i}^{\perp} \boldsymbol{x}$. We then have

$$\mathbb{P}\left(\text{sign}(\langle \boldsymbol{x}, \boldsymbol{\mu}_I \rangle) \neq \text{sign}(\langle \boldsymbol{x}, \widehat{\boldsymbol{\theta}} \rangle) \middle| I = i, \widehat{I} \neq i\right) = \mathbb{P}\left(\text{sign}(\langle \boldsymbol{x}, \boldsymbol{\mu}_i \rangle)\left(\langle \boldsymbol{x}, \boldsymbol{\mu}_i \rangle \theta + \langle \mathcal{P}_{\boldsymbol{\mu}_i}^{\perp} \boldsymbol{x}, \mathcal{P}_{\boldsymbol{\mu}_i}^{\perp} \widehat{\boldsymbol{\theta}} \rangle\right) \leq 0 \middle| I = i, \widehat{I} \neq i\right).$$

Note that $\boldsymbol{x}$ is a test data point, independent of the training data $\mathcal{T}$ and so it is independent of $\widehat{\boldsymbol{\theta}}$. In addition, under our data model, $\langle \boldsymbol{x}, \boldsymbol{\mu}_i \rangle$ is independent of $\mathcal{P}_{\boldsymbol{\mu}_i}^{\perp} \boldsymbol{x}$. Hence, $\text{sign}(\langle \boldsymbol{x}, \boldsymbol{\mu}_i \rangle)\langle \mathcal{P}_{\boldsymbol{\mu}_i}^{\perp} \boldsymbol{x}, \mathcal{P}_{\boldsymbol{\mu}_i}^{\perp} \widehat{\boldsymbol{\theta}} \rangle \sim \mathcal{N}(0, \|\mathcal{P}_{\boldsymbol{\mu}_i}^{\perp} \widehat{\boldsymbol{\theta}}\|_{\ell_2}^2)$. Given that $\|\widehat{\boldsymbol{\theta}}\|_{\ell_2} = 1$ we also have $\|\mathcal{P}_{\boldsymbol{\mu}_i}^{\perp} \widehat{\boldsymbol{\theta}}\|_{\ell_2}^2 = 1 - \langle \widehat{\boldsymbol{\theta}}, \boldsymbol{\mu}_i \rangle^2 = 1 - \theta^2$. In short, we can write

$$\text{sign}(\langle \boldsymbol{x}, \boldsymbol{\mu}_i \rangle)\langle \mathcal{P}_{\boldsymbol{\mu}_i}^{\perp} \boldsymbol{x}, \mathcal{P}_{\boldsymbol{\mu}_i}^{\perp} \widehat{\boldsymbol{\theta}} \rangle = \sqrt{1 - \theta^2} Z, \quad Z \sim \mathcal{N}(0, 1).$$

Using this characterization, we proceed by writing

$$\mathbb{P}\left(\text{sign}(\langle \boldsymbol{x}, \boldsymbol{\mu}_I \rangle) \neq \text{sign}(\langle \boldsymbol{x}, \widehat{\boldsymbol{\theta}} \rangle) \middle| I = i, \widehat{I} \neq i\right) = \mathbb{P}\left(\text{sign}(\langle \boldsymbol{x}, \boldsymbol{\mu}_i \rangle)\langle \boldsymbol{x}, \boldsymbol{\mu}_i \rangle \theta + \sqrt{1 - \theta^2} Z \leq 0\right)$$

$$= \mathbb{P}\left(Z \leq -\frac{|\langle \boldsymbol{x}, \boldsymbol{\mu}_i \rangle| \theta}{\sqrt{1 - \theta^2}}\right)$$

$$= 1 - \mathbb{E}\left[\Phi\left(\frac{\theta}{\sqrt{1 - \theta^2}} |\langle \boldsymbol{x}, \boldsymbol{\mu}_i \rangle|\right)\right]$$

$$\stackrel{(a)}{\geq} 1 - \mathbb{E}\left[\Phi\left(\sqrt{\frac{1 + \rho}{1 - \rho}} |\langle \boldsymbol{x}, \boldsymbol{\mu}_i \rangle|\right)\right]$$

$$\stackrel{(b)}{\geq} 1 - \Phi\left(\sqrt{\frac{1 + \rho}{1 - \rho}} \mathbb{E}[|\langle \boldsymbol{x}, \boldsymbol{\mu}_i \rangle|]\right)$$

$$\stackrel{(c)}{\geq} 1 - \Phi\left(2\sqrt{\frac{1 + \rho}{1 - \rho}}\right) = \Phi\left(-2\sqrt{\frac{1 + \rho}{1 - \rho}}\right),$$

where $(a)$ follows from the fact that $\theta \leq \sqrt{\frac{1+\rho}{2}}$; $(b)$ holds due to Jensen's inequality and concavity of $\Phi(\cdot)$ on the positive values, and $(c)$ holds because under our data model $\mathbb{E}[|\langle \boldsymbol{x}, \boldsymbol{\mu}_i \rangle|] = \mathbb{E}[y(1 + u)\|\boldsymbol{\mu}_i\|_{\ell_2}^2] = \mathbb{E}[1 + u] \leq 2$ since $\|u\|_{\psi_2} = 1$. This completes the proof of claim.

**Proof of Lemma D.2.** Define $q_i = p_i - 1/k$. Note that $q_i$ can be negative, and we have $\sum_{i=1}^{k} q_i = 0$. We write

$$
\begin{aligned}
H(p_1, \ldots, p_k) &= -\sum_{i=1}^{k} p_i \log p_i \\
&= -\sum_{i=1}^{k} (1/k + q_i) \log(1/k + q_i) \\
&= -\sum_{i=1}^{k} (1/k + q_i)[\log(1/k) + \log(1 + kq_i)] \\
&\geq \log k - \sum_{i=1}^{k} (1/k + q_i) kq_i \\
&= \log k - k \sum_{i=1}^{k} q_i^2 \\
&= \log k - k \sum_{i=1}^{k} (p_i - 1/k)^2 \, .
\end{aligned}
\tag{D.10}
$$

Note that in Equation (D.10) we used the fact that $1 + kq_i \geq 0$ and $\log x \leq x - 1$ for all $x \geq 0$.

This completes the proof of the lemma.

**Proof of Lemma D.3.** Define a $\rho$-cover of $\mathbb{S}^{d-1}$ as a set of $\mathcal{V} \coloneqq \{v_1, \ldots, v_N\}$ such that for any $\theta \in \mathbb{S}^{d-1}$, there exists some $v_i$ such that $\langle \theta, v_i \rangle \geq \rho$. The $\rho$-covering number of $\mathbb{S}^{d-1}$ is

$$
N(\rho, \mathbb{S}^{d-1}) \coloneqq \inf\{N \in \mathbb{N} : \text{there exists a } \rho\text{-cover } \mathcal{V} \text{ of } \mathbb{S}^{d-1} \text{ with size } N\} \, .
$$

By a simple argument we have $M(\rho, \mathbb{S}^{d-1}) \geq N(\rho, \mathbb{S}^{d-1})$. Concretely, we construct a $\rho$-packing greedily by adding an element at each step which has inner product at most $\rho$ with all the previously selected elements, until it is no longer possible. This means that any point on $\mathbb{S}^{d-1}$ has inner product larger than $\rho$ by some of the elements in the constructed set (otherwise it contradicts its maximality). Hence, we have a set that is both a $\rho$-cover and a $\rho$-packing of $\mathbb{S}^{d-1}$, and by definition it results in $M(\rho, \mathbb{S}^{d-1}) \geq N(\rho, \mathbb{S}^{d-1})$.

We next lower bound $N(\rho, \mathbb{S}^{d-1})$ via a volumetric argument. Let $\mathcal{V} \coloneqq \{v_1, \ldots, v_N\}$ be a $\rho$-cover of $\mathbb{S}^{d-1}$. For each element $v_i \in \mathcal{V}$ we consider the cone around it with apex angle $\cos^{-1}(\rho)$. Its intersection with $\mathbb{S}^{d-1}$ defines a spherical cap which we denote by $\mathcal{C}(v_i, \rho)$. Since $\mathcal{V}$ forms a $\rho$-cover of $\mathbb{S}^{d-1}$, we have

$$
\text{Vol}(\mathbb{S}^{d-1}) \leq \text{Vol}(\cup_{i=1}^{N} \mathcal{C}(v_i, \rho)) \leq \sum_{i=1}^{N} \text{Vol}(\mathcal{C}(v_i, \rho)) \, .
$$

We next use Lemma 2.2 from Ball et al. (1997) by which we have $\frac{\mathcal{C}(v_i, \rho)}{\text{Vol}(\mathbb{S}^{d-1})} \leq e^{-d\rho^2/2}$. Using this above, we get

$$
1 \leq N e^{-d\rho^2/2}
$$

for any $\rho$-cover $\mathcal{V}$. Thus, we have $N(\rho, \mathbb{S}^{d-1}) \geq \exp(\frac{d\rho^2}{2})$, which completes the proof of the lemma.

# E    Comparison with Other Lower Bounds for Learning with Noisy Labels

Cai & Wei (2021); Im & Grigas (2023); Zhu et al. (2024) derive some noteworthy lower bounds in much more general settings than ours; naturally, their bounds are weaker than our bound in Theorem 4.6. In the work of Cai & Wei (2021), our setting corresponds to $n_p = 0, n_q = n$. As per Theorem 3.2 of Cai & Wei (2021), the lower bound on the error is effectively[9] $n^{-O(\frac{1+\alpha}{d})}$. So when $\alpha \ll d$, this lower bound yields a much worse sample complexity than our result in Theorem 4.6. In Im & Grigas (2023), the lower bound on the error (Theorem 2) does not reduce with $n$, so even if there are infinite samples, we cannot get zero error in the worst case. As for Zhu et al. (2024), their lower bound on the error (Theorem 1) also has a non-diminishing term depending on $\epsilon$. In the special case of $\epsilon = 0$ (or $\epsilon$ being small enough), there is an $n^{-1/2}$ dependence but no dependence on the dimension or a related quantity. However, the upper bound in Theorem 2 therein does have a dependence on a VC dimension-like quantity as expected, so their lower bound is probably loose w.r.t. dimension.

---

[9]This is because $\beta \leq 1$ as per Definition 2 and $\alpha\beta \leq d$ as per the paragraph after Remark 3 (of Cai & Wei (2021)).

# F   Proof of Theorem 4.8

Similar to the proof of Theorem 4.1, we want to upper bound $\mathbb{P}(y\langle \boldsymbol{x}, \widehat{\boldsymbol{\theta}}_1 \rangle < 0)$.[10] Plugging in $\widehat{\boldsymbol{\theta}}_1$ from (4.8) we have

$$y\langle \boldsymbol{x}, \widehat{\boldsymbol{\theta}}_1 \rangle = \frac{y}{n}\langle \boldsymbol{x}, \sum_i \widetilde{y}_i \boldsymbol{x}_i \rangle = \frac{y}{n}\langle \boldsymbol{x}, \sum_i \text{sign}(\langle \boldsymbol{x}_i, \widehat{\boldsymbol{\theta}}_0 \rangle)\boldsymbol{x}_i \rangle$$

A major complication is that $\widehat{\boldsymbol{\theta}}_0$ depends on all the data points in the training set. Expanding $\widehat{\boldsymbol{\theta}}_0$, we get

$$\widehat{\boldsymbol{\theta}}_0 = \frac{1}{n}\sum_i \widehat{y}_i \boldsymbol{x}_i = \frac{1}{n}\sum_i \xi_i y_i \boldsymbol{x}_i = \frac{1}{n}\left( (\sum_i \xi_i(1+u_i))\boldsymbol{\mu} + \boldsymbol{\Sigma}^{1/2}\sum_i \xi_i y_i \boldsymbol{z}_i \right). \tag{F.1}$$

Thus,

$$\widetilde{y}_\ell = \text{sign}(\langle \boldsymbol{x}_\ell, \widehat{\boldsymbol{\theta}}_0 \rangle) = \text{sign}\left( \langle \boldsymbol{x}_\ell, (\sum_i \xi_i(1+u_i))\boldsymbol{\mu} + \boldsymbol{\Sigma}^{1/2}\sum_i \xi_i y_i \boldsymbol{z}_i \rangle \right). \tag{F.2}$$

Here is the outline of our proof:

- For every $\ell \in [n]$ we define the dummy label

$$\widetilde{y}'_\ell = \text{sign}\left( \langle \boldsymbol{x}_\ell, (\sum_i \xi_i(1+u_i))\boldsymbol{\mu} + \boldsymbol{\Sigma}^{1/2}\xi_\ell y_\ell \boldsymbol{z}_\ell \rangle \right). \tag{F.3}$$

  Note that $\widetilde{y}'_\ell$ depends only on $\boldsymbol{z}_\ell$, whereas $\widetilde{y}_\ell$ depends on all $\{\boldsymbol{z}_i\}_{i\in[n]}$.

- We define the event $\mathcal{E} := \{\forall \ell : \widetilde{y}'_\ell = \text{sign}(\langle \boldsymbol{x}_\ell, \widehat{\boldsymbol{\theta}}_0 \rangle)\}$.

- We have

$$\mathbb{P}(y\langle \boldsymbol{x}, \widehat{\boldsymbol{\theta}}_1 \rangle < 0) = \mathbb{P}(y\langle \boldsymbol{x}, \widehat{\boldsymbol{\theta}}_1 \rangle < 0; \mathcal{E}) + \mathbb{P}(y\langle \boldsymbol{x}, \widehat{\boldsymbol{\theta}}_1 \rangle < 0; \mathcal{E}^c)$$
$$\leq \mathbb{P}(y\langle \boldsymbol{x}, \widehat{\boldsymbol{\theta}}_1 \rangle < 0; \mathcal{E}) + \mathbb{P}(\mathcal{E}^c). \tag{F.4}$$

- We bound each of the term on the right-hand side separately. Note that under the event $\mathcal{E}$, we have

$$y\langle \boldsymbol{x}, \widehat{\boldsymbol{\theta}}_1 \rangle = \frac{y}{n}\langle \boldsymbol{x}, \sum_\ell \widetilde{y}'_\ell \boldsymbol{x}_\ell \rangle.$$

  We will control the probability of the RHS above being negative using concentration results on the sum of i.i.d. subgaussian random variables (with appropriate conditioning first).

Next we get into details of this proof outline. We start by bounding $\mathbb{P}(\mathcal{E}^c)$.

**Lemma F.1.** *Fix $\ell \in [n]$. Suppose that $nd \geq \frac{\gamma^4}{\lambda_{\max}^2}$. Define $p' := (1 + \frac{3\gamma^4}{8\lambda_{\max}^2 nd})p$. We then have*

$$\mathbb{P}\left(\widetilde{y}'_\ell \neq \text{sign}(\langle \boldsymbol{x}_\ell, \widehat{\boldsymbol{\theta}}_0 \rangle)\right) = \mathbb{P}\left(\widetilde{y}'_\ell \langle \boldsymbol{x}_\ell, \widehat{\boldsymbol{\theta}}_0 \rangle < 0\right) \leq \frac{1}{2}\left( \exp\left(-\frac{\gamma^4}{8\lambda_{\max}^2}(1-2p')\frac{n}{d}\right) + e^{-d/16} \right)e^{d/n}.$$

Using Lemma F.1 along with union bounding over $\ell \in [n]$ (note that the events are dependent), we get

$$\mathbb{P}(\mathcal{E}^c) \leq \frac{n}{2}\left( \exp\left(-\frac{\gamma^4}{8\lambda_{\max}^2}(1-2p')\frac{n}{d}\right) + e^{-d/16} \right)e^{d/n}, \tag{F.5}$$

where $p' = (1 + \frac{3\gamma^4}{8\lambda_{\max}^2 nd})p$.

---

[10]Note that $\mathbb{P}(\text{sign}(\langle \boldsymbol{x}, \widehat{\boldsymbol{\theta}}_1 \rangle) \neq y) = \mathbb{P}(y\langle \boldsymbol{x}, \widehat{\boldsymbol{\theta}}_1 \rangle < 0)$.

Next we note that under the event $\mathcal{E}$, we have:

$$y\langle \boldsymbol{x}, \widehat{\boldsymbol{\theta}}_1 \rangle = \frac{y}{n}\langle \boldsymbol{x}, \sum_\ell \widetilde{y}_\ell' \boldsymbol{x}_\ell \rangle = \frac{1}{n}\sum_{\ell \in [n]} y\,\mathrm{sign}\left(\langle \boldsymbol{x}_\ell, (\sum_i \xi_i(1+u_i))\boldsymbol{\mu} + \boldsymbol{\Sigma}^{1/2}\xi_\ell y_\ell \boldsymbol{z}_\ell \rangle\right)\langle \boldsymbol{x}, \boldsymbol{x}_\ell \rangle. \tag{F.6}$$

We define the shorthand $\beta := \sum_i \xi_i(1+u_i)$ and condition on $\beta$. Then (F.6) will be sum of i.i.d. subgaussian variables

$$T_\ell := y\,\mathrm{sign}\left(\langle \boldsymbol{x}_\ell, \beta\boldsymbol{\mu} + \boldsymbol{\Sigma}^{1/2}\xi_\ell y_\ell \boldsymbol{z}_\ell \rangle\right)\langle \boldsymbol{x}, \boldsymbol{x}_\ell \rangle, \quad \ell \in [n]. \tag{F.7}$$

We continue by first characterizing its expectation.

**Lemma F.2.** *Suppose that $\frac{\beta\gamma^2}{2\lambda_{\max}d} > 1$ and $d \geq 7$. Then for $\ell \in [n]$, we have*

$$\mathbb{E}\left[T_\ell | \beta\right] \geq (1-2q)\langle y\boldsymbol{x}, \boldsymbol{\mu} \rangle > 0,$$

*where $q := \exp\left(-\frac{\beta\gamma^2}{20\lambda_{\max}}\right)$.*

Recall that $\boldsymbol{x}_\ell = y_\ell(1+u_\ell)\boldsymbol{\mu} + \boldsymbol{\Sigma}^{1/2}\boldsymbol{z}_\ell$ and therefore

$$\begin{aligned}
\|T_\ell - \mathbb{E}[T_\ell|\beta]\|_{\psi_2} \leq 2\|T_\ell\|_{\psi_2} &= 2\|\langle \boldsymbol{x}, \boldsymbol{x}_\ell \rangle\|_{\psi_2} \\
&= 2\|\langle \boldsymbol{x}, y_\ell(1+u_\ell)\boldsymbol{\mu} + \boldsymbol{\Sigma}^{1/2}\boldsymbol{z}_\ell \rangle\|_{\psi_2} \\
&\leq 4|\langle \boldsymbol{x}, \boldsymbol{\mu} \rangle| + 2\left\|\boldsymbol{\Sigma}^{1/2}\boldsymbol{x}\right\|_{\ell_2},
\end{aligned}$$

where we used the assumption $\|u_\ell\|_{\psi_2} = 1$ and the fact that $\|\boldsymbol{z}\|_{\psi_2} = 1$.

Next by using Hoeffding-type inequality for sum of sub-gaussian random variables (see e.g., Proposition 5.1 of Vershynin (2010)) we have for every $t \geq 0$,

$$\mathbb{P}\left(\left|\sum_\ell T_\ell - \sum_\ell \mathbb{E}[T_\ell|\beta]\right| \geq t \Big| \beta\right) \leq 2\exp\left(-\frac{t^2}{64n(\langle \boldsymbol{x}, \boldsymbol{\mu} \rangle^2 + \left\|\boldsymbol{\Sigma}^{1/2}\boldsymbol{x}\right\|_{\ell_2}^2)}\right).$$

Therefore,

$$\begin{aligned}
\mathbb{P}(\sum_\ell T_\ell < 0 \,|\beta) &\leq \mathbb{P}\left(|\sum_\ell T_\ell - \sum_\ell \mathbb{E}[T_\ell|\beta]| \geq \sum_\ell \mathbb{E}[T_\ell|\beta] \,\Big|\, \beta\right) \\
&\leq 2\exp\left(-\frac{n(\mathbb{E}[T_\ell|\beta])^2}{64(\langle \boldsymbol{x}, \boldsymbol{\mu} \rangle^2 + \left\|\boldsymbol{\Sigma}^{1/2}\boldsymbol{x}\right\|_{\ell_2}^2)}\right) \\
&\leq 2\exp\left(-\frac{n(1-2q)^2\langle y\boldsymbol{x}, \boldsymbol{\mu} \rangle^2}{64(\langle \boldsymbol{x}, \boldsymbol{\mu} \rangle^2 + \left\|\boldsymbol{\Sigma}^{1/2}\boldsymbol{x}\right\|_{\ell_2}^2)}\right),
\end{aligned} \tag{F.8}$$

where the last step follows from Lemma F.2. Recall that $q = \exp\left(-\frac{\beta\gamma^2}{20\lambda_{\max}}\right)$.

Our next step is to take expectation of the above with respect to $\beta = \sum_i \xi_i(1+u_i)$. Before proceeding we define the following event:

$$\mathcal{E}_0 := \{\beta \geq n(1-2p)/2\}. \tag{F.9}$$

Note that by Hoeffding-type inequality for sub-gaussian random variables and given that $\mathbb{E}[\xi_i] = 1 - 2p$, we have

$$\mathbb{P}\left(\left|\sum_i \xi_i(1+u_i) - n(1-2p)(1+\mathbb{E}[u_i])\right| \geq t\right) \leq 2\exp\left(-\frac{t^2}{8n}\right).$$

By choosing $t = n(1-2p)/2$, and using the fact that $u_i > 0$ for all $i$, we get

$$\mathbb{P}\left(\sum_i \xi_i(1+u_i) < n(1-2p)/2\right) \leq 2\exp\left(-\frac{n}{32}(1-2p)^2\right).$$

Hence,

$$\mathbb{P}(\mathcal{E}_0^c) \le 2\exp\left(-\frac{n}{32}(1-2p)^2\right).$$

We now continue by taking expectation of (F.8) with respect to $\beta$.

$$\mathbb{P}(\sum_\ell T_\ell < 0) \le 2\,\mathbb{E}\left[\exp\left(-\frac{n(1-2q)^2\langle y\boldsymbol{x}, \boldsymbol{\mu}\rangle^2}{64(\langle \boldsymbol{x}, \boldsymbol{\mu}\rangle^2 + \|\boldsymbol{\Sigma}^{1/2}\boldsymbol{x}\|_{\ell_2}^2)}\right)\right]$$

$$= 2\,\mathbb{E}\left[\exp\left(-\frac{n(1-2q)^2\langle y\boldsymbol{x}, \boldsymbol{\mu}\rangle^2}{64(\langle \boldsymbol{x}, \boldsymbol{\mu}\rangle^2 + \|\boldsymbol{\Sigma}^{1/2}\boldsymbol{x}\|_{\ell_2}^2)}\right)(\mathbb{1}_{\mathcal{E}_0} + \mathbb{1}_{\mathcal{E}_0^c})\right]$$

$$\le 2\,\mathbb{E}\left[\exp\left(-\frac{n(1-2q)^2\langle y\boldsymbol{x}, \boldsymbol{\mu}\rangle^2}{64(\langle \boldsymbol{x}, \boldsymbol{\mu}\rangle^2 + \|\boldsymbol{\Sigma}^{1/2}\boldsymbol{x}\|_{\ell_2}^2)}\right)\mathbb{1}_{\mathcal{E}_0}\right] + 2\mathbb{P}(\mathcal{E}_0^c). \tag{F.10}$$

On the event $\mathcal{E}_0$ we have

$$\frac{\beta\gamma^2}{2\lambda_{\max}d} \ge \frac{n(1-2p)\gamma^2}{4\lambda_{\max}d} > 1,$$

by our requirement in the theorem statement. Therefore,

$$q = \exp\left(-\frac{\beta\gamma^2}{20\lambda_{\max}}\right) \le \exp\left(-\frac{n(1-2p)\gamma^2}{40\lambda_{\max}}\right) := q'.$$

Since $1 - 2q \ge 1 - 2q' > 0$ this implies that

$$\mathbb{E}\left[\exp\left(-\frac{n(1-2q)^2\langle y\boldsymbol{x}, \boldsymbol{\mu}\rangle^2}{64(\langle \boldsymbol{x}, \boldsymbol{\mu}\rangle^2 + \|\boldsymbol{\Sigma}^{1/2}\boldsymbol{x}\|_{\ell_2}^2)}\right)\mathbb{1}_{\mathcal{E}_0}\right] \le \mathbb{E}\left[\exp\left(-\frac{n(1-2q')^2\langle y\boldsymbol{x}, \boldsymbol{\mu}\rangle^2}{64(\langle \boldsymbol{x}, \boldsymbol{\mu}\rangle^2 + \|\boldsymbol{\Sigma}^{1/2}\boldsymbol{x}\|_{\ell_2}^2)}\right)\mathbb{1}_{\mathcal{E}_0}\right]$$

$$\le \exp\left(-\frac{n(1-2q')^2\langle y\boldsymbol{x}, \boldsymbol{\mu}\rangle^2}{64(\langle \boldsymbol{x}, \boldsymbol{\mu}\rangle^2 + \|\boldsymbol{\Sigma}^{1/2}\boldsymbol{x}\|_{\ell_2}^2)}\right).$$

Using the above bound along with the bound on $\mathbb{P}(\mathcal{E}_0^c)$ into (F.10), we arrive at

$$\mathbb{P}(\sum_\ell T_\ell < 0) \le 2\exp\left(-\frac{n(1-2q')^2\langle y\boldsymbol{x}, \boldsymbol{\mu}\rangle^2}{64(\langle \boldsymbol{x}, \boldsymbol{\mu}\rangle^2 + \|\boldsymbol{\Sigma}^{1/2}\boldsymbol{x}\|_{\ell_2}^2)}\right) + 4\exp\left(-\frac{n}{32}(1-2p)^2\right). \tag{F.11}$$

Invoking (F.6) and the definition of $T_\ell$ given by (F.7) we obtain

$$\mathbb{P}(y\langle \boldsymbol{x}, \widehat{\boldsymbol{\theta}}_1\rangle < 0; \mathcal{E}) \le 2\exp\left(-\frac{n(1-2q')^2\langle y\boldsymbol{x}, \boldsymbol{\mu}\rangle^2}{64(\langle \boldsymbol{x}, \boldsymbol{\mu}\rangle^2 + \|\boldsymbol{\Sigma}^{1/2}\boldsymbol{x}\|_{\ell_2}^2)}\right) + 4\exp\left(-\frac{n}{32}(1-2p)^2\right). \tag{F.12}$$

Finally we combine (F.12), (F.5) and (F.4) to get

$$\mathbb{P}(y\langle \boldsymbol{x}, \widehat{\boldsymbol{\theta}}_1\rangle < 0) \le 2\exp\left(-\frac{n(1-2q')^2\langle y\boldsymbol{x}, \boldsymbol{\mu}\rangle^2}{64(\langle \boldsymbol{x}, \boldsymbol{\mu}\rangle^2 + \|\boldsymbol{\Sigma}^{1/2}\boldsymbol{x}\|_{\ell_2}^2)}\right)$$

$$+ 4\exp\left(-\frac{n}{32}(1-2p)^2\right) + \frac{n}{2}\left(\exp\left(-\frac{\gamma^4}{8\lambda_{\max}^2}(1-2p')\frac{n}{d}\right) + e^{-d/16}\right)e^{d/n}. \tag{F.13}$$

This finishes the proof of Theorem 4.8.

We will now prove the intermediate lemmas F.1 and F.2.

## F.1 Proof of Lemma F.1

First note that $\mathbb{P}\left(\widetilde{y}_\ell' \neq \text{sign}(\langle \boldsymbol{x}_\ell, \widehat{\boldsymbol{\theta}}_0\rangle)\right) = \mathbb{P}\left(\widetilde{y}_\ell'\langle \boldsymbol{x}_\ell, \widehat{\boldsymbol{\theta}}_0\rangle < 0\right)$.

Define $\tilde{\xi}_i := \xi_i(1 + u_i)$. Then using (F.1), we have

$$\widetilde{y}_\ell'\langle \boldsymbol{x}_\ell, \widehat{\boldsymbol{\theta}}_0\rangle = \frac{\widetilde{y}_\ell'}{n}\langle \boldsymbol{x}_\ell, (\sum_i \tilde{\xi}_i)\boldsymbol{\mu} + \boldsymbol{\Sigma}^{1/2}\xi_\ell y_\ell \boldsymbol{z}_\ell + \boldsymbol{\Sigma}^{1/2}\sum_{i\neq\ell}\xi_i y_i \boldsymbol{z}_i\rangle.$$

We have $\sum_{i\neq\ell}\xi_i y_i \boldsymbol{z}_i \sim \mathcal{N}(\boldsymbol{0}, (n-1)\boldsymbol{I}_d)$ and so $\langle \widetilde{y}_\ell'\boldsymbol{x}_\ell, \boldsymbol{\Sigma}^{1/2}\sum_{i\neq\ell}\xi_i y_i \boldsymbol{z}_i\rangle$ is a zero mean Gaussian with variance $(n-1)\left\|\boldsymbol{\Sigma}^{1/2}\boldsymbol{x}_\ell\right\|_{\ell_2}^2$. Let $\mathcal{F}_\ell$ be the $\sigma$-algebra generated by $(\boldsymbol{z}_\ell, y_\ell, \{\xi_i, u_i\}_{i\in[n]})$. Note that $\boldsymbol{z}_i$, for $i \neq \ell$, is independent of $\mathcal{F}_\ell$ and $\widetilde{y}_\ell'$ is $\mathcal{F}_\ell$-measurable. Hence,

$$\mathbb{P}\left(\widetilde{y}_\ell'\langle \boldsymbol{x}_\ell, \widehat{\boldsymbol{\theta}}_0\rangle < 0 \big| \mathcal{F}_\ell\right) = \Phi\left(-\frac{\widetilde{y}_\ell'\langle \boldsymbol{x}_\ell, (\sum_i \tilde{\xi}_i)\boldsymbol{\mu} + \boldsymbol{\Sigma}^{1/2}\xi_\ell y_\ell \boldsymbol{z}_\ell\rangle}{\sqrt{n-1}\left\|\boldsymbol{\Sigma}^{1/2}\boldsymbol{x}_\ell\right\|_{\ell_2}}\right)$$

$$\overset{(a)}{=} \Phi\left(-\frac{\left|\langle \boldsymbol{x}_\ell, (\sum_i \tilde{\xi}_i)\boldsymbol{\mu} + \boldsymbol{\Sigma}^{1/2}\xi_\ell y_\ell \boldsymbol{z}_\ell\rangle\right|}{\sqrt{n-1}\left\|\boldsymbol{\Sigma}^{1/2}\boldsymbol{x}_\ell\right\|_{\ell_2}}\right)$$

$$\overset{(b)}{\leq} \frac{1}{2}\exp\left(-\frac{\left|\langle \boldsymbol{x}_\ell, (\sum_i \tilde{\xi}_i)\boldsymbol{\mu} + \boldsymbol{\Sigma}^{1/2}\xi_\ell y_\ell \boldsymbol{z}_\ell\rangle\right|^2}{2n\left\|\boldsymbol{\Sigma}^{1/2}\boldsymbol{x}_\ell\right\|_{\ell_2}^2}\right)$$

$$\overset{(c)}{\leq} \frac{1}{2}\exp\left(-\frac{\langle \boldsymbol{x}_\ell, \boldsymbol{\mu}\rangle^2(\sum_i \tilde{\xi}_i)^2}{4n\left\|\boldsymbol{\Sigma}^{1/2}\boldsymbol{x}_\ell\right\|_{\ell_2}^2}\right)\exp\left(\frac{\langle \boldsymbol{x}_\ell, \boldsymbol{\Sigma}^{1/2}\boldsymbol{z}_\ell\rangle^2}{2n\left\|\boldsymbol{\Sigma}^{1/2}\boldsymbol{x}_\ell\right\|_{\ell_2}^2}\right)$$

$$\leq \frac{1}{2}\exp\left(-\frac{\langle \boldsymbol{x}_\ell, \boldsymbol{\mu}\rangle^2(\sum_i \tilde{\xi}_i)^2}{4n\left\|\boldsymbol{\Sigma}^{1/2}\boldsymbol{x}_\ell\right\|_{\ell_2}^2}\right)\exp\left(\frac{\|\boldsymbol{z}_\ell\|_{\ell_2}^2}{2n}\right)$$

$$\overset{(d)}{\leq} \frac{1}{2}\exp\left(-\frac{\gamma^4(\sum_i \tilde{\xi}_i)^2}{4n\lambda_{\max}^2\|\boldsymbol{z}_\ell\|_{\ell_2}^2}\right)\exp\left(\frac{\|\boldsymbol{z}_\ell\|_{\ell_2}^2}{2n}\right). \tag{F.14}$$

Here, $(a)$ follows by the choice of $\widetilde{y}_\ell'$; $(b)$ holds by using Equation (7.1.13) of Abramowitz & Stegun (1968):

$$\Phi^c(t) \leq \sqrt{\frac{2}{\pi}}\frac{e^{-t^2/2}}{t + \sqrt{t^2 + \frac{8}{\pi}}} \leq \frac{1}{2}e^{-t^2/2},$$

for all $t \geq 0$. In addition $(c)$ holds since $(a + b)^2 \geq \frac{a^2}{2} - b^2$. Also, $(d)$ holds because under our data model we have $\boldsymbol{\Sigma}^{1/2}\boldsymbol{x}_\ell = \boldsymbol{\Sigma}\boldsymbol{z}_\ell$ and so $\left\|\boldsymbol{\Sigma}^{1/2}\boldsymbol{x}_\ell\right\|_{\ell_2} \leq \lambda_{\max}\|\boldsymbol{z}_\ell\|_{\ell_2}$. Further, $|\langle \boldsymbol{x}_\ell, \boldsymbol{\mu}\rangle| = |y(1 + u_\ell)\|\boldsymbol{\mu}\|_{\ell_2}^2| \geq \gamma^2$ because $u_\ell \geq 0$. Putting these bounds together, we have

$$\frac{\langle \boldsymbol{x}_\ell, \boldsymbol{\mu}\rangle}{\left\|\boldsymbol{\Sigma}^{1/2}\boldsymbol{x}_\ell\right\|_{\ell_2}} \geq \frac{\gamma^2}{\lambda_{\max}\|\boldsymbol{z}_\ell\|_{\ell_2}}.$$

We proceed by taking expectation of the right-hand side of (F.14) with respect to $\tilde{\xi}_1, \ldots, \tilde{\xi}_n$. Define the shorthand $\lambda := \frac{\gamma^4}{4n\lambda_{\max}^2\|\boldsymbol{z}_\ell\|_{\ell_2}^2}$. Fix an arbitrary $\lambda_0 > 0$ for now (we will determine its value later) and define truncated parameter

$\bar{\lambda} = \min(\lambda, \lambda_0)$. We have

$$\mathbb{E}\left[e^{-\lambda(\sum_i \tilde{\xi}_i)^2}\right] \le \mathbb{E}\left[e^{-\bar{\lambda}(\sum_i \tilde{\xi}_i)^2}\right] \tag{F.15}$$

$$= \mathbb{E}\left[e^{-\bar{\lambda}\sum_i(1+u_i)^2} e^{-\bar{\lambda}\sum_{i\neq j}\xi_i\xi_j(1+u_i)(1+u_j)}\right]$$

$$\le e^{-\bar{\lambda}n} \prod_{i\neq j} \mathbb{E}[e^{-\bar{\lambda}\xi_i(1+u_i)}]\,\mathbb{E}[e^{-\bar{\lambda}\xi_j(1+u_j)}]$$

$$= e^{-\bar{\lambda}n} (\mathbb{E}[e^{-\bar{\lambda}\xi(1+u)}])^{n(n-1)}$$

$$= e^{-\bar{\lambda}n} (\mathbb{E}[(1-p)e^{-\bar{\lambda}(1+u)} + pe^{\bar{\lambda}(1+u)}])^{n(n-1)}$$

$$\le e^{-\bar{\lambda}n} \left((1-p)e^{-\bar{\lambda}} + pe^{\bar{\lambda}}\,\mathbb{E}[e^{\bar{\lambda}u}]\right)^{n(n-1)}$$

$$\le e^{-\bar{\lambda}n} \left((1-p)e^{-\bar{\lambda}} + pe^{\bar{\lambda}}e^{\bar{\lambda}^2/2}\right)^{n(n-1)}$$

$$= e^{-\bar{\lambda}n^2} \left(1 - p + pe^{2\bar{\lambda}+\bar{\lambda}^2/2}\right)^{n(n-1)}. \tag{F.16}$$

Here, the first and the second inequality follows from $u_i \ge 0$. The third inequality holds since $u$ has unit sub-gaussian norm and so it moment-generating function is bounded as $\mathbb{E}[e^{\lambda u}] \le e^{\lambda^2/2}$.

We next further upper bound the right-hand side of F.16 to get a simpler expression. In doing that we use the following lemma (proved in Appendix F.1.1).

**Lemma F.3.** *For $t \le \frac{1}{2}$, we have $e^t - 1 \le t + t^2$.*

By choosing $\lambda_0 \le 0.2$ we have $2\lambda_0 + \lambda_0^2/2 \le 1/2$ and so $2\bar{\lambda} + \bar{\lambda}^2/2 \le 1/2$. Using the above lemma, we have

$$e^{2\bar{\lambda}+\bar{\lambda}^2/2} - 1 \le 2\bar{\lambda} + \bar{\lambda}^2/2 + \left(2\bar{\lambda} + \bar{\lambda}^2/2\right)^2$$

$$= 2\bar{\lambda} + \bar{\lambda}^2/2 + 4\bar{\lambda}^2 + \bar{\lambda}^4/4 + 2\bar{\lambda}^3$$

$$\le 2\bar{\lambda} + \bar{\lambda}\left(\lambda_0/2 + 4\lambda_0 + \lambda_0^3/4 + 2\lambda_0^2\right)$$

$$\le (2 + 6\lambda_0)\bar{\lambda},$$

where we used $\lambda_0 \le 0.2$ in the last step. Next using the inequality $1 + x \le e^x$ for $x \ge 0$, we get

$$(e^{2\bar{\lambda}+\bar{\lambda}^2/2} - 1)p + 1 \le (2 + 6\lambda_0)p\bar{\lambda} + 1 \le e^{(2+6\lambda_0)\bar{\lambda}p}.$$

By using this bound in (F.16) we obtain

$$\mathbb{E}\left[e^{-\lambda(\sum_i \tilde{\xi}_i)^2}\right] \le e^{-\bar{\lambda}n^2} e^{(2+6\lambda_0)\bar{\lambda}pn^2} = e^{-(1-2p')\bar{\lambda}n^2}, \tag{F.17}$$

with $p' := (1 + 3\lambda_0)p$.

Using the above in (F.14) and then taking expectation with respect to $\boldsymbol{z}_\ell$ we get

$$\mathbb{P}\left(\widetilde{y}'_\ell\langle \boldsymbol{x}_\ell, \widehat{\boldsymbol{\theta}}_0\rangle < 0\right) \le \frac{1}{2}\,\mathbb{E}\left[\exp\left(-(1-2p')\bar{\lambda}n^2\right)\exp\left(\frac{\|\boldsymbol{z}_\ell\|_{\ell_2}^2}{2n}\right)\right]$$

$$\le \frac{1}{2}\,\mathbb{E}\left[\exp\left(-2(1-2p')\bar{\lambda}n^2\right)\right]^{1/2}\mathbb{E}\left[\exp\left(\frac{\|\boldsymbol{z}_\ell\|_{\ell_2}^2}{n}\right)\right]^{1/2}, \tag{F.18}$$

by applying the Cauchy–Schwarz inequality.

Using the moment generating function of $\chi_d^2$ distribution, we have

$$\mathbb{E}\left[\exp\left(\frac{\|\boldsymbol{z}_\ell\|_{\ell_2}^2}{n}\right)\right] = \left(1 - \frac{2}{n}\right)^{-d/2} \le e^{2d/n}, \tag{F.19}$$

for $n \geq 4$. Here, we use the inequality $(1 - x)^{-1} \leq e^{2x}$ for $x \in [0, 1/2]$.

We continue by bounding the first term on the right-hand side of (F.18). Define the event $\mathcal{E}_1$ as follows:

$$\mathcal{E}_1 := \left\{ \|\boldsymbol{z}_\ell\|_{\ell_2} \leq \sqrt{2d} \right\} . \tag{F.20}$$

We then have

$$
\begin{aligned}
\mathbb{E}\left[\exp\left(-2(1-2p')\bar{\lambda}n^2\right)\right] &= \mathbb{E}\left[\exp\left(-2(1-2p')\bar{\lambda}n^2\right)\left(\mathbb{1}_{\mathcal{E}_1} + \mathbb{1}_{\mathcal{E}_1^c}\right)\right] \\
&\leq \mathbb{E}\left[\exp\left(-2(1-2p')\bar{\lambda}n^2\right)\mathbb{1}_{\mathcal{E}_1}\right] + \mathbb{E}\left[\mathbb{1}_{\mathcal{E}_1^c}\right] \\
&\overset{(a)}{=} \mathbb{E}\left[\exp\left(-\frac{\gamma^4}{4\lambda_{\max}^2 d}(1-2p')n\right)\mathbb{1}_{\mathcal{E}_1}\right] + \mathbb{P}(\mathcal{E}_1^c) \\
&\leq \exp\left(-\frac{\gamma^4}{4\lambda_{\max}^2}(1-2p')\frac{n}{d}\right) + \mathbb{P}(\mathcal{E}_1^c)
\end{aligned}
\tag{F.21}
$$

Note that in step $(a)$, we used the fact that on the event $\mathcal{E}_1$, we have $\lambda \geq \gamma^4/(8\lambda_{\max}^2 nd)$ and we choose $\lambda_0 = \gamma^4/(8\lambda_{\max}^2 nd)$. This way, we have $\bar{\lambda} = \min(\lambda, \lambda_0) = \gamma^4/(8\lambda_{\max}^2 nd)$. Also note that by our assumption in the statement of the lemma, we have $\lambda_0 \leq 0.2$ which is the condition assumed in deriving (F.17).

We next bound $\mathbb{P}(\mathcal{E}_1^c)$. Since $\boldsymbol{z}_\ell \sim \mathcal{N}(0, \boldsymbol{I}_d)$, $\|\boldsymbol{z}_\ell\|_{\ell_2}^2$ has $\chi^2$ distribution with $d$-degrees of freedom. Using the tail bound of $\chi^2$ distribution (see Theorem 3 of Ghosh (2021)) we have $\mathbb{P}(\|\boldsymbol{z}_\ell\|_{\ell_2}^2 \geq 2d) \leq e^{-d/8}$ and so

$$\mathbb{P}(\mathcal{E}_1^c) \leq \mathbb{P}(\|\boldsymbol{z}_\ell\|_{\ell_2}^2 \geq 2d) \leq e^{-d/8}.$$

Using the above in (F.21) we obtain

$$\mathbb{E}\left[\exp\left(-2(1-2p')\bar{\lambda}n^2\right)\right] \leq \exp\left(-\frac{\gamma^4}{4\lambda_{\max}^2}(1-2p')\frac{n}{d}\right) + e^{-d/8}. \tag{F.22}$$

By combining (F.22) and (F.19) with (F.18) we arrive at

$$
\begin{aligned}
\mathbb{P}\left(\widetilde{y}_\ell'\langle\boldsymbol{x}_\ell, \widehat{\boldsymbol{\theta}}_0\rangle < 0\right) &\leq \frac{1}{2}\left(\exp\left(-\frac{\gamma^4}{4\lambda_{\max}^2}(1-2p')\frac{n}{d}\right) + e^{-d/8}\right)^{1/2} e^{d/n} \\
&\leq \frac{1}{2}\left(\exp\left(-\frac{\gamma^4}{8\lambda_{\max}^2}(1-2p')\frac{n}{d}\right) + e^{-d/16}\right) e^{d/n},
\end{aligned}
$$

which completes the proof of lemma.

### F.1.1 PROOF OF LEMMA F.3

Using the Taylor series expansion of $e^t$, we have:

$$e^t - 1 = t + \frac{t^2}{2} + \sum_{j=3}^{\infty} \frac{t^j}{j!} \tag{F.23}$$

$$\leq t + \frac{t^2}{2} + \frac{t^2}{2}\sum_{j=3}^{\infty} \frac{1}{j!} \tag{F.24}$$

$$= t + \frac{t^2}{2} + \frac{t^2}{2}\left(e - \frac{5}{2}\right) \tag{F.25}$$

(F.24) follows by using the fact that for $j \geq 3$ and $t \leq \frac{1}{2}$, $t^j \leq t^3 \leq \frac{t^2}{2}$. (F.25) follows by using the fact that $e^1 - 1 = 1 + \frac{1}{2} + \sum_{j=3}^{\infty} \frac{1}{j!}$ after plugging in $t = 1$ in Equation (F.23).

Next, using the fact that $e < 3$ in (F.25), we get:

$$e^t - 1 < t + \frac{3t^2}{4} < t + t^2. \tag{F.26}$$

This completes the proof.

## F.2 Proof of Lemma F.2

Under our data model we have $\boldsymbol{x}_\ell = y_\ell(1 + u_\ell)\boldsymbol{\mu} + \boldsymbol{\Sigma}^{1/2}\boldsymbol{z}_\ell$. Substituting for $\boldsymbol{x}_\ell$ in the expression of $T_\ell$, we get

$$
\begin{aligned}
T_\ell &= \text{sign}\left(\langle y_\ell(1 + u_\ell)\boldsymbol{\mu} + \boldsymbol{\Sigma}^{1/2}\boldsymbol{z}_\ell, \beta\boldsymbol{\mu} + \xi_\ell y_\ell\boldsymbol{\Sigma}^{1/2}\boldsymbol{z}_\ell\rangle\right)\langle y\boldsymbol{x}, y_\ell(1 + u_\ell)\boldsymbol{\mu} + \boldsymbol{\Sigma}^{1/2}\boldsymbol{z}_\ell\rangle \\
&= \text{sign}\left(y_\ell(1 + u_\ell)\beta\gamma^2 + \xi_\ell y_\ell\left\|\boldsymbol{\Sigma}^{1/2}\boldsymbol{z}_\ell\right\|_{\ell_2}^2\right)\langle y\boldsymbol{x}, y_\ell(1 + u_\ell)\boldsymbol{\mu} + \boldsymbol{\Sigma}^{1/2}\boldsymbol{z}_\ell\rangle,
\end{aligned}
$$

using the fact that $\boldsymbol{\Sigma}^{1/2}\boldsymbol{\mu} = 0$. Taking expectation we get

$$
\mathbb{E}[T_\ell|\beta] = \mathbb{E}\left[\text{sign}\left(y_\ell(1 + u_\ell)\beta\gamma^2 + \xi_\ell y_\ell\left\|\boldsymbol{\Sigma}^{1/2}\boldsymbol{z}_\ell\right\|_{\ell_2}^2\right)\langle y\boldsymbol{x}, y_\ell(1 + u_\ell)\boldsymbol{\mu}\rangle\,\Big|\,\beta\right]
$$

since the other term will be an odd function of $\boldsymbol{z}_\ell$. Letting

$$
q_0 := \mathbb{P}\left((1 + u_\ell)\beta\gamma^2 + \xi_\ell\left\|\boldsymbol{\Sigma}^{1/2}\boldsymbol{z}_\ell\right\|_{\ell_2}^2 < 0\right),
$$

we get

$$
\mathbb{E}[T_\ell|\beta] = (1 - 2q_0)\langle y\boldsymbol{x}, (1 + u_\ell)\boldsymbol{\mu}\rangle \geq (1 - 2q_0)\langle y\boldsymbol{x}, \boldsymbol{\mu}\rangle, \tag{F.27}
$$

because $u_\ell > 0$ and $\langle y\boldsymbol{x}, \boldsymbol{\mu}\rangle > 0$, given the positive margin in our data model, and if $q_0 \leq \frac{1}{2}$. So what remains is to show that $q_0 \leq q \leq 1/2$.

We write

$$
\begin{aligned}
q_0 &\leq \mathbb{P}\left((1 + u_\ell)\beta\gamma^2 - \lambda_{\max}\|\boldsymbol{z}_\ell\|_{\ell_2}^2 < 0\right) \\
&\leq \mathbb{P}\left(\frac{\beta\gamma^2}{\lambda_{\max}} < \|\boldsymbol{z}_\ell\|_{\ell_2}^2\right) \qquad\qquad \text{(since } u_\ell > 0) \\
&< \exp\left(-\frac{\beta\gamma^2}{20\lambda_{\max}}\right),
\end{aligned}
$$

if $\frac{\beta\gamma^2}{2d\lambda_{\max}} > 1$, where the last step follows from the observation that $\|\boldsymbol{z}_\ell\|_{\ell_2}^2 \sim \chi_d^2$ and using the tail bound of $\chi_d^2$ (Theorem 3 of Ghosh (2021)). So when $\frac{\beta\gamma^2}{2d\lambda_{\max}} > 1$ and $d \geq 7$,

$$
q_0 \leq q := \exp\left(-\frac{\beta\gamma^2}{20\lambda_{\max}}\right) \leq e^{-d/10} < \frac{1}{2}.
$$

Plugging this into (F.27) completes the proof.

## G  Proof of Theorem 4.9

Using Theorem 4.8 (and because the training set $\mathcal{T}$ has measure zero), we have $\text{err}(\widehat{\boldsymbol{\theta}}_1) \leq \mathbb{E}[\alpha_1(\boldsymbol{x})]$. Based on our data model, $\boldsymbol{x} = y(1 + u)\boldsymbol{\mu} + \boldsymbol{\Sigma}^{1/2}\boldsymbol{z}$. Consider the probabilistic event $\mathcal{E}_2 := \{\boldsymbol{z} : \|\boldsymbol{z}\|_{\ell_2}^2 \leq 2d\}$. For every $\eta > 0$, we have

$$
\begin{aligned}
\mathbb{E}\left[\exp\left(-\frac{\eta\langle\boldsymbol{x}, \boldsymbol{\mu}\rangle^2}{\langle\boldsymbol{x}, \boldsymbol{\mu}\rangle^2 + \left\|\boldsymbol{\Sigma}^{1/2}\boldsymbol{x}\right\|_{\ell_2}^2}\right)\right] &= \mathbb{E}\left[\exp\left(-\frac{\eta\langle\boldsymbol{x}, \boldsymbol{\mu}\rangle^2}{\langle\boldsymbol{x}, \boldsymbol{\mu}\rangle^2 + \left\|\boldsymbol{\Sigma}^{1/2}\boldsymbol{x}\right\|_{\ell_2}^2}\right)(\mathbb{1}_{\mathcal{E}_2} + \mathbb{1}_{\mathcal{E}_2^c})\right] \\
&\leq \exp\left(-\frac{\eta\gamma^4}{\gamma^4 + 2\lambda_{\max}^2 d}\right) + \mathbb{P}(\mathcal{E}_2^c),
\end{aligned}
$$

where the inequality holds because $\langle\boldsymbol{x}, \boldsymbol{\mu}\rangle = y(1 + u)\gamma^2$ and so $\langle\boldsymbol{x}, \boldsymbol{\mu}\rangle^2 \geq \gamma^4$ (since $u$ is a non-negative random variable). In addition,

$$
\left\|\boldsymbol{\Sigma}^{1/2}\boldsymbol{x}\right\|_{\ell_2}^2 = \|\boldsymbol{\Sigma}\boldsymbol{z}\|_{\ell_2}^2 \leq \lambda_{\max}^2\|\boldsymbol{z}\|_{\ell_2}^2 \leq 2\lambda_{\max}^2 d,
$$

under the event $\mathcal{E}_2$. Further, $\|z\|_{\ell_2}^2 \sim \chi_d^2$ and so using tail bounds for $\chi_d^2$ (see Theorem 3 of Ghosh (2021)), we have $\mathbb{P}(\mathcal{E}_2^c) \leq e^{-d/8}$. Putting things together, we obtain

$$\mathbb{E}\left[\exp\left(-\frac{\eta\langle\boldsymbol{x},\boldsymbol{\mu}\rangle^2}{\langle\boldsymbol{x},\boldsymbol{\mu}\rangle^2 + \left\|\boldsymbol{\Sigma}^{1/2}\boldsymbol{x}\right\|_{\ell_2}^2}\right)\right] \leq \exp\left(-\frac{\eta\gamma^4}{\gamma^4 + 2\lambda_{\max}^2 d}\right) + e^{-d/8}.$$

The claim of theorem follows by setting $\eta = \frac{n}{64}(1 - 2q')^2$.

## H  Remaining Experimental Details

Here we provide the remaining details about the experiments in Section 5. Our experiments were done using TensorFlow and JAX. In all the cases, we retrain starting from random initialization rather than the previous checkpoint we converged to before RT; the former worked better than the latter. We list training details for each individual dataset next.

**CIFAR-10.**  Optimizer is SGD with momentum = 0.9, batch size = 32, number of epochs in each stage of training (i.e., both stages of baseline, full RT and consensus-based RT) = 30. We use the cosine one-cycle learning rate schedule with initial learning rate = 0.1 for each stage of training. The number of epochs and initial learning rate were chosen based on the performance of the baseline method and *not* based on the performance of full or consensus-based RT. Standard augmentations such as random cropping, flipping and brightness/contrast change were used.

**CIFAR-100.**  Details are the same as CIFAR-10 except that here the number of epochs in each stage of training = 40 and initial learning rate = 0.005.

**DomainNet.**  Details are the same as CIFAR-10 except that here the number of epochs in each stage of training = 15, initial learning rates for the first stage of the baseline in Tables 7, 8, and 9 are $1e-3$, $5e-2$, and $5e-4$, respectively, whereas the initial learning rate for the second stage of the baseline as well as full RT and consensus-based RT is $1e-3$.

**AG News Subset.**  Small BERT model link: `https://www.kaggle.com/models/tensorflow/bert/frameworks/tensorFlow2/variations/bert-en-uncased-l-4-h-512-a-8/versions/2?tfhub-redirect=true`, BERT English uncased preprocessor link: `https://www.kaggle.com/models/tensorflow/bert/frameworks/tensorFlow2/variations/en-uncased-preprocess/versions/3?tfhub-redirect=true`. Optimizer is Adam with fixed learning rate = $1e-5$, batch size = 32, number of epochs in each training stage = 5.

## I  Accuracy over the Entire Dataset and over the Consensus Set in the Case of AG News Subset

Here we list the accuracies of the predicted labels and the given labels over the entire dataset and the accuracy of the predicted labels (= given labels) over the consensus set for AG News Subset. Just like in Table 3, the accuracy of the predicted labels over the consensus set is significantly more than the accuracy of the predicted and given labels over the entire dataset. This explains why consensus-based RT performs the best, even though the consensus set is much smaller than the full dataset (~ 28%, 32% and 38% of the entire training set for the three values of $\epsilon$).

*Table 11.* **AG News Subset.** Accuracies of predicted labels and given labels over the entire dataset and accuracies of predicted labels (= given labels) over the consensus set. Conclusions are the same as in Table 3.

| $\epsilon$ | Acc. of *predicted* labels on *full* dataset | Acc. of *given* labels on *full* dataset | Acc. of *predicted* labels on *consensus* set |
|---|---|---|---|
| 0.3 | $53.20 \pm 2.82$ | $32.52 \pm 2.05$ | $\mathbf{61.81} \pm 2.66$ |
| 0.5 | $66.78 \pm 1.31$ | $35.5 \pm 0.14$ | $\mathbf{76.48} \pm 0.93$ |
| 0.8 | $79.98 \pm 0.80$ | $42.53 \pm 0.13$ | $\mathbf{89.59} \pm 0.43$ |

## J  Consensus-Based Retraining Does Better than Confidence-Based Retraining

Here we compare full and consensus-based RT against another strategy for retraining which we call **confidence-based retraining (RT)**. Specifically, we propose to retrain with the predicted labels of the samples with the top 50% margin (i.e., highest predicted probability - second highest predicted probability); margin is a measure of the model's confidence. This idea is similar to self-training's method of sample selection in the semi-supervised setting (Amini et al., 2022). In Tables 12 and 13, we show results for CIFAR-10 and CIFAR-100 (in the same setting as Section 5 and Appendix H) with the smallest value of $\epsilon$ from Tables 1 and 2, respectively. Notice that *consensus-based RT is clearly better than confidence-based RT*.

*Table 12.* **CIFAR-10.** Test set accuracies (mean ± standard deviation). *Consensus-based RT performs the best.*

| $\epsilon$ | Baseline | Full RT | Consensus-based RT | Confidence-based RT |
|---|---|---|---|---|
| 1 | 57.78 ± 1.13 | 60.07 ± 0.63 | **63.84** ± 0.56 | 62.09 ± 0.55 |

*Table 13.* **CIFAR-100.** Test set accuracies (mean ± standard deviation). Again, *consensus-based RT performs the best.*

| $\epsilon$ | Baseline | Full RT | Consensus-based RT | Confidence-based RT |
|---|---|---|---|---|
| 3 | 23.53 ± 1.01 | 24.42 ± 1.22 | **29.98** ± 1.11 | 24.99 ± 1.25 |

## K  Retraining is Beneficial Even Without a Validation Set

In all our previous experiments, we assumed access to a small clean validation set. Here we show the results of training ResNet-34 on CIFAR-100 when we do *not* have access to a validation set and train for 100 epochs, and also compare them against the corresponding results with a validation set where we trained for 40 epochs (Table 4 in Section 5). The point of these experiments is to show that retraining can offer gains even without a validation set, in which case we train for too many epochs and expect overfitting. Please see the results and discussion in Table 14.

*Table 14.* **CIFAR-100 w/ ResNet-34 without (top) and with (bottom) a validation set.** Test set accuracies (mean ± standard deviation). Note that retraining (in particular, consensus-based RT) leads to improvement even without a validation set. However, the performance and amount of improvement obtained with retraining are worse in the absence of a validation set due to a higher degree of overfitting. This is not surprising.

| | $\epsilon$ | Baseline | Full RT | Consensus-based RT |
|---|---|---|---|---|
| **Without** Val. Set | 4 | 26.03 ± 1.75 | 28.33 ± 1.76 | **30.47** ± 1.31 |
| | 5 | 42.50 ± 1.14 | 44.43 ± 1.27 | **46.27** ± 1.72 |

| | $\epsilon$ | Baseline | Full RT | Consensus-based RT |
|---|---|---|---|---|
| **With** Val. Set | 4 | 37.53 ± 1.58 | 39.33 ± 1.37 | **43.87** ± 1.62 |
| (Same as Table 4) | 5 | 51.13 ± 0.69 | 52.33 ± 0.38 | **55.43** ± 0.42 |

## L  Beyond Label DP: Evaluating Retraining in the Presence of Human Annotation Errors

Even though our empirical focus in this paper has been label DP training, retraining (RT) can be employed for general problems with label noise. Here we evaluate RT in a setting with "real" label noise due to *human annotation*. Specifically, we focus on training a ResNet-18 model (*without* label DP to be clear) on the CIFAR-100N dataset introduced by Wei et al. (2022) and available on the TensorFlow website. CIFAR-100N is just CIFAR-100 labeled by humans; thus, it has real human annotation errors. The experimental setup and details are the same as CIFAR-100 (as stated in Section 5 and Appendix H); the only difference is that here we use initial learning rate = 0.01.

In Table 15, we list the test accuracies of the baseline which is just standard training with the given labels, full RT and consensus-based RT, respectively. Even here *with human annotation errors*, consensus-based RT is beneficial.

*Table 15.* **CIFAR-100N.** Test set accuracies (mean ± standard deviation). *So even with real human annotation errors, consensus-based RT improves performance.*

| Baseline | Full RT | Consensus-based RT |
|---|---|---|
| 55.47 ± 0.18 | 56.88 ± 0.35 | **57.68** ± 0.35 |

