# OpenReview forum: "Retraining with Predicted Hard Labels Provably Increases Model Accuracy"
_ICML.cc/2025/Conference — ICML 2025 poster_

### Official Review · Reviewer_JtmD · 2025-03-07

**Overall Recommendation:** 3

**Summary:**

This paper investigates the benefits of retraining a model using its own predicted hard labels in scenarios where training data contains noisy labels.

There are two strategies for retraining the model:

- *Full Retraining:* The model is retrained on the entire dataset using its own predicted hard labels.
- *Consensus-Based Retraining:* Only samples for which the model's predicted label matches the original noisy label are used for retraining.

The paper provides a rigorous theoretical analysis showing that *full retraining* with predicted hard labels can improve a model's population accuracy. In a linearly separable binary classification setting with randomly flipped labels, the authors derive error bounds and sufficient conditions when retraining is beneficial.

The authors also conduct extensive experiments on datasets such as CIFAR-10, CIFAR-100, and AG News Subset (a language dataset). The results show that both full retraining and consensus-based retraining enhance model performance, with consensus-based retraining providing the most significant improvements.

## update after rebuttal

The detailed response has resolved my concerns. Thus, I raise my score after the rebuttal.

**Claims And Evidence:**

The paper supports its main claims with a combination of rigorous theoretical analysis and extensive empirical validation.

However, there are some aspects where the evidence is less complete:

- The theoretical analysis focuses on full retraining under a uniform label noise model, while the consensus-based retraining, which empirically shows superior performance, lacks a corresponding theoretical analysis.
- The experiments are conducted on moderate-scale datasets, so the scalability and generalizability of the approach to larger or more complex settings (e.g., experiments on the ImageNet dataset) remain to be further explored.

**Essential References Not Discussed:**

All essential related works are cited or discussed in the paper.

**Experimental Designs Or Analyses:**

The benchmark datasets are widely recognized in the community, and the experimental designs are reasonable. But some experimental settings remain unclear (see questions).

**Methods And Evaluation Criteria:**

The proposed methods, namely full retraining and consensus-based retraining, make sense to tackle the challenges of learning with noisy labels and label differential privacy. Additionally, benchmark datasets like CIFAR-10, CIFAR-100, CIFAR-100N, and AG News Subset are well widely recognized.

**Other Comments Or Suggestions:**

The meaning of $\epsilon$ is not explained in the introduction, yet it appears in both the abstract and the conclusion of the introduction. Readers who are not familiar with Label Differential Privacy (DP) may be confused. It would be beneficial to provide an intuitive explanation of $\epsilon$ in the introduction to enhance clarity and accessibility.

**Other Strengths And Weaknesses:**

This paper investigates the benefits of retraining a model using its own predicted hard labels for label differential privacy (DP) and provides theoretical analysis. However, it exists several limitations.

First, the theoretical analysis is confined to binary classification using linear models. Consequently, the derived results and error bounds are limited in scope and may not extend to practical scenarios where many tasks involve multiclass classification and complex nonlinear models. In real-world applications, sufficiently powerful nonlinear models can potentially memorize all the noisy labels. As a result, the model's predicted hard labels would simply replicate the noisy labels, rendering full retraining ineffective. This limitation suggests that while the theoretical contributions are valuable for understanding retraining in controlled settings, their applicability to more complex, realistic models remains questionable.

Second, although consensus-based retraining shows superior performance empirically, the paper does not provide a corresponding theoretical framework to analyze its behavior or guarantees.

Third, the effectiveness of retraining is heavily dependent on the accuracy of the initial model's predictions. In scenarios where the initial model performs poorly, the retraining process might not yield significant improvements.

**Questions For Authors:**

1. In scenarios where the initial model has low accuracy, how does the retraining process behave? Addressing this question could clarify the robustness of your method and whether it remains effective when the initial model is weak.
2. Regarding the training details (Lines 1439–1440), why must the number of gradient steps and the initial learning rate be chosen based on the performance of the baseline method? Are the retraining methods particularly sensitive to these hyperparameters?
3. What are the noise rates corresponding to different values of $\epsilon$?
4. The authors explicitly state that the forward correction algorithm is applied in the initial training stage for the experiments in Table 5. However, what loss function is used in the initial training stage for the experiments in Tables 1, 2, 4, and 6? Is it the standard cross-entropy loss? Clarifying this would improve the reproducibility of the reported results.

**Relation To Broader Scientific Literature:**

The key contributions of the paper relate to two broader scientific literatures:

- **Learning with Noisy Labels:** There is a lot of work on training models in the presence of noisy labels, which often involves robust loss functions or noise-correction techniques. The paper contributes to this literature by offering the first theoretical guarantees showing that full retraining with predicted hard labels can provably improve model accuracy under uniform label noise.
- **Label Differential Privacy (DP):** In the context of privacy-preserving machine learning, label DP has emerged as an important concept. Prior works have proposed various noise-injection mechanisms (such as randomized response) to ensure privacy for sensitive label information. This paper shows that retraining methods (full retraining and consensus-based retraining) can enhance the model's performance without additional privacy costs.

**Theoretical Claims:**

Main theorems (Theorem 4.1, Theorem 4.2, Theorem 4.8, Theorem 4.9) are checked.

---

> ### Author Rebuttal · Authors · 2025-04-01
>
> Thanks for your review and questions! We address your concerns below.
>
> **Other Strengths And Weaknesses:**
>
> **1. "First, the theoretical analysis is…remains questionable."**:
> * We agree that our analysis on linear models for binary classification will not fully explain what happens in the case of non-linear models for multi-class classification, and we don’t intend to oversell the scope of our theoretical results. But we believe it is valuable as a first step; after all, *ours is the first work to analyze retraining with hard labels in any setting*. Moreover, we believe that some of our proof ideas could be useful even in the analysis of non-linear models. For instance, the proof technique of constructing dummy predicted labels that match the actual predicted labels with high probability (see lines 307-319 left column) should be useful in general, because the issue of dependence of each predicted label on the entire training set is universal regardless of the model type.
>
> * Regarding your point about complex models perfectly fitting noisy labels, we completely agree. And that is why, for such expressive models, it is important to apply (both in theory and practice) some kind of regularization when training them with noisy labels; for e.g., $\ell_2$ regularization, early stopping, etc. Applying regularization is reasonable in scenarios such as label DP, where we already know that the labels will be noisy.
>
> **2. "Second, although consensus-based retraining…its behavior or guarantees." / first bullet point under Claims And Evidence**: Agreed. We have admitted this limitation in Section 6, and plan to analyze consensus-based retraining in the future. Please note that the analysis of full retraining is itself pretty non-trivial (main technical challenges have been discussed after Thm. 4.8) and interesting in our opinion.
>
> We do acknowledge the above two weaknesses. However, it is usually very difficult to perfectly align theoretical analysis with practical settings, and it is common to analyze simplified settings. So we believe these weaknesses **do not fundamentally undermine the significance of our work**.
>
> **3. "Third, the effectiveness of retraining…yield significant improvements."** Indeed, retraining should intuitively only be beneficial when *the initial model’s predictions are more accurate than the given (noisy) labels* used to train the initial model. We have discussed/demonstrated this in several parts of the paper – Fig. 1 (see its caption), Tables 3 and 7 (these are on real data), and the comment on the range of $n$ after Remark 4.10 (specifically, regarding the lower bound on $n$). Moreover, in Appendix J & Table 10, we did an *ablation study* with and without a validation set. The initial model is naturally weaker w/o a validation set (due to overfitting); despite this, *retraining is still beneficial* but the gains are less than those with a val set. This observation is not surprising.
>
> **Second bullet point under Claims And Evidence:** As mentioned in Section 6, performing larger experiments is left for future work. While we didn’t have the time to train ImageNet from scratch now, we ran experiments on *DomainNet* dataset (available in Tensorflow) which has 345 classes & is much larger than CIFAR. We did linear probing (due to lack of time) with features extracted from a ResNet-50 pretrained on ImageNet. The setup is similar to our full fine-tuning experiments. DomainNet Results:
>
> |$\epsilon$|Baseline|Full RT|Consensus-based RT|
> |---|---|---|---|
> |$3$|$23.60\pm0.92$|$29.23\pm1.03$|$\mathbf{36.30}\pm0.75$|
> |$4$|$48.25\pm0.05$|$52.10\pm0.10$|$\mathbf{57.40}\pm0.20$|
>
> *So even here RT (especially, consensus RT) yields large gains*.
>
> **Questions For Authors:**
>
> **1.** Please see the response to weakness **3** above (especially the last two sentences about the ablation).
>
> **2.** They *need not* be chosen based on the baseline’s performance. We did this to avoid any further hyper-parameter tuning based on retraining – to demonstrate that retraining is *not* very sensitive to hyper-parameters. If one were to optimize the hyper-parameters based on retraining’s performance as well, the gains would only increase.
>
> **3.** If randomized response (RR) is used as the baseline, then with $C$ classes and for $\epsilon$-labelDP, each sample receives its true label $y$ w.p. $\frac{e^{\epsilon}}{e^{\epsilon} + C-1}$ and some other label $y'$ w.p. $\frac{1}{e^{\epsilon} + C-1}$ for all $y' \neq y$ (this has been explained in lines 177-181 left column). If the method of Ghazi et al. (2021) is used, then their first stage is RR (so the same as before), but the noise level of subsequent stages depends on the performance of the previous stage's model.
>
> **4.** Standard cross-entropy loss was used; we’ll mention this in the next version. Thanks for pointing this out!
>
> We hope to have resolved your concerns and we're happy to discuss further. If you’re satisfied, *we sincerely hope you will raise your score*!

---

> > ### Comment · Reviewer_JtmD · 2025-04-02
> >
> > Thanks for your detailed rebuttal and extra experiments. My concerns have been resolved. Then, I decide to raise my recommendation score.

---

> > > ### Author Response · Authors · 2025-04-04
> > >
> > > Thanks for raising your score! We’ll add the extra experiments (and important clarifications from the rebuttal) in the next version.

---

### Official Review · Reviewer_hYpN · 2025-03-10

**Overall Recommendation:** 4

**Summary:**

The authors theoretically analyze retraining in a linearly separable binary classification problem and show that it can improve the model accuracy with respect to the initial training in presence of label noise. They show that retraining is particularly helpful with high levels of label noise. Then, the paper empirically shows that the proposed consensus-based retraining works better than the normal retraining.

## Update after rebuttal
After reading all the reviews carefully and considering the additional effort made by the authors, I decided to raise my score from 3 to 4. I think this is an excellent paper.

**Claims And Evidence:**

The claims are almost all clear and convincing.
- The main claim for which the clarity could be improved is the specification (especially in the abstract) that they theoretically analyze a **binary** classification problem.
- In line 123 you claim that your work is on the fully supervised setting. Isn't the label noise scenario considered weakly-supervised?

**Essential References Not Discussed:**

I am not aware of important related work that is not cited in the paper.

**Experimental Designs Or Analyses:**

I would have preferred to see a comparison with other algorithms that perform classification with label noise, but I only see a minor result on the combination of retraining and forward correction. I think that a wider comparison would help in understanding if the contribution of this paper is mainly theoretical or if there is also a possible advancement for state-of-the-art techniques.
I don't understand why the authors did not share the code. This arises concerns on the reproducibility of their results.

**Methods And Evaluation Criteria:**

The proposed evaluation criteria make sense for the problem considered.

**Other Comments Or Suggestions:**

- in line 165 at the beginning of pp. 4 I would prefer the authors to use $\cdot$ instead of $.$

**Other Strengths And Weaknesses:**

Strenghts
- The paper is well written and even though it is theoretically heavy, it can be easily read by non-experts
- The related work section is very useful
- The experimental results enforces the theoretical claims

Weaknesses
- No code
- No comparison with other techniques for classification with label noise (apart from Forward correction)
- No theoretical analysis or comments for the multi-class classification problem

**Questions For Authors:**

- How do you relate the retraining technique with the problem of memorization of noisy samples? That is a well known problem in the noisy labels literature and I am afraid that retraining could worsen the memorization effect. Can you provide an empirical analysis of the memorization effect when using retraining? [a,b,c]

- What does it happen when we increase the number of gradient steps? Does the gap between the accuracy achieved with and without retraining decrease? Is there a point in which, if we train the model for X steps, the retraining lowers the accuracy? Maybe this would be an interesting ablation study.

- You use the baseline in Ghazi et al. 2021. Which objective function do you use to train your neural networks? I assume you use the cross-entropy. However, you did not study how the performance of your algorithm would change by changing baseline or objective function. This could arise questions on the general validity of retraining. Could you study the performance applying these changes?


[a] Arpit, D., Jastrzębski, S., Ballas, N., Krueger, D., Bengio, E., Kanwal, M. S., ... & Lacoste-Julien, S. (2017, July). A closer look at memorization in deep networks. In International conference on machine learning (pp. 233-242). PMLR.

[b] Zhang, C., Bengio, S., Hardt, M., Recht, B., & Vinyals, O. (2016). Understanding deep learning requires rethinking generalization. ICLR 2017.

[c]  Liu, S., Niles-Weed, J., Razavian, N., & Fernandez-Granda, C. (2020). Early-learning regularization prevents memorization of noisy labels. Advances in neural information processing systems, 33, 20331-20342.

**Relation To Broader Scientific Literature:**

The paper contributions are incremental, as the retraining technique is well known. However, the theoretical analysis is interesting and novel in my opinion.

**Theoretical Claims:**

I checked the theoretical claims superficially and they seem correct and well written.

---

> ### Author Rebuttal · Authors · 2025-04-01
>
> Thanks for your review and great questions! We address your questions/concerns below.
>
> **Claims And Evidence:**
> 1. We will clarify "binary" in the abstract.
> 2. Here we simply meant a setting where we have labels for all samples - to distinguish it from the setting of self-training where we are *not* given labels for all the samples. We’ll clarify this.
>
> **Experimental Designs Or Analyses / first two weaknesses**:
>
> * Regarding comparisons with other noise-robust methods, please note that we are *not* claiming retraining is a SOTA *general-purpose* noise-robust method (see lines 100-103 left column). We are just advocating it as **straightforward post-processing step** that can be applied **on top of vanilla training or a noise-robust training method**. In case it wasn’t clear, Table 5 shows results wherein initial training (baseline) was done with forward correction applied to the method of Ghazi et al. 2021, and retraining was done on top of this. Please also see our response to your third question (under Questions For Authors) below, where we show that retraining is beneficial as a post-processing step even *when using a noise-robust loss function* instead of the usual cross-entropy loss. Moreover, it’s not straightforward to apply many existing noise-robust methods to sophisticated label DP mechanisms (such as Ghazi et al.); retraining is very easy to apply in contrast.
>
> * We didn’t release the code because at the time of submission, we didn’t obtain our organization's approval to release it. We weren't sure if code can be shared in the rebuttal because the email on rebuttal instructions didn't mention anything about code. We will release the code upon paper acceptance.
>
> **No theoretical analysis or comments for the multi-class case (Weakness 3):** Extending our analysis to the multi-class case is left for future work. Here is a starting point: in the case of $C$ classes, the labels $y_i$’s will be $C$-dimensional one-hot vectors, the ground truth $\Theta^{\ast}$ will be a $C \times d$ matrix (features $x_i$’s are still $d$-dimensional vectors, but $y_i$’s need to be defined appropriately in terms of $\Theta^{\ast}$ and the $x_i$’s) and our predictor $\hat{\Theta} = \frac{1}{n} \sum_i y_i x_i^T$ will also be in $\mathbb{R}^{C \times d}$.
>
> **Questions For Authors:**
>
> **1.** Yes, memorization of noisy labels with powerful models is an issue. And if initial training is done naively, retraining may exacerbate this issue. That is why in almost all our experiments (except in Appendix J), we assume access to a clean validation set; please also see footnote 5 for the *practical version of this assumption*. This prevents the model from heavily memorizing. Moreover, as we show in Tables 3 & 7, the accuracy of the predicted (= given) labels on the consensus set is much more than the accuracy of both the predicted and given labels on the full set. This shows that regulated initial training is effective at avoiding memorization. Further, *as shown in Appendix J, even in the absence of a validation set, retraining is still beneficial but the gains are less – this is expected because the initial model’s performance is degraded due to more memorization/overfitting here*.
>
> **2.** Indeed, the benefit of retraining decreases when initial training is done for a larger number of steps. We studied this in Appendix J – here we don’t have a validation set and trained blindly for 100 epochs. Due to more overfitting here, the gains of retraining are lower than the corresponding gains with a validation set where we stopped at 40 epochs. If we train for even longer, the initial model will heavily memorize the noisy labels and this will probably render retraining ineffective.
>
> **3.** Yes, we used the cross-entropy (CE) loss; we’ll state this in the next version. Our baseline for AG News is actually randomized response (see lines 425-426 left column) to demonstrate the generality of retraining w.r.t. the baseline. Further, in Table 5, our baseline is forward correction applied to the method of Ghazi et al. 2021. So we do have results with other baselines. And based on your suggestion, we performed experiments with the *noise-robust symmetric CE loss function* proposed in [1] (1k+ citations) instead of the vanilla CE loss. In their loss (eq. 7 of [1]), we set $\alpha=0.8$ and $\beta=0.2$. Here are the results for CIFAR-100 w/ ResNet-34.
>
> |$\epsilon$|Baseline|Full RT|Consensus-based RT|
> |---|---|---|---|
> |$4$|$37.07\pm2.03$|$38.17\pm2.03$|$\mathbf{43.20}\pm1.77$|
> |$5$|$53.10\pm0.54$|$53.40\pm0.33$|$\mathbf{56.13}\pm0.25$|
>
> Thus, consensus RT yields meaningful gains even with the loss function of [1].
>
> We hope to have resolved your concerns and we are happy to discuss further. If you’re satisfied with our answers, *we sincerely hope you will raise your score*!
>
> ----
>
> [1]: Wang, Yisen, et al. "Symmetric cross entropy for robust learning with noisy labels." ICCV 2019.

---

> > ### Comment · Reviewer_hYpN · 2025-04-02
> >
> > Thank you for the answers. I will keep my score as it is.

---

> > > ### Author Response · Authors · 2025-04-07
> > >
> > > Thanks for your reply.
> > >
> > > We are adding some new results on a *bigger dataset to show that retraining is effective when applied on top of label noise correcting methods*. Specifically, we show results when the baseline is *forward correction* and *backward correction* (from Patrini et al. 2017 cited in the paper) applied to the first stage of Ghazi et al. 2021 (similar to Table 5 in the paper); these results are in (A) and (B) below, respectively. For comparison in (C) below, we also show results when the baseline is just Ghazi et al. 2021 (i.e., no correction is applied). These results are on the *DomainNet* dataset (available on Tensorflow) *which has 345 classes and is much larger than CIFAR*. We did linear probing (using cross-entropy loss) with features extracted from a ResNet-50 pretrained on ImageNet. The setup is similar to our full fine-tuning experiments.
> > >
> > > **(A) Baseline = Forward Correction (Patrini et al. 2017) + Ghazi et al. 2021:**
> > > |$\epsilon$|Baseline|Full RT|Consensus-based RT|
> > > |---|---|---|---|
> > > |$3$|$31.23\pm0.56$|$33.30\pm0.65$|$\mathbf{36.07}\pm0.78$|
> > > |$4$|$58.50\pm0.08$|$58.63\pm0.12$|$\mathbf{61.80}\pm0.08$|
> > >
> > > **(B) Baseline = Backward Correction (Patrini et al. 2017) + Ghazi et al. 2021:**
> > > |$\epsilon$|Baseline|Full RT|Consensus-based RT|
> > > |---|---|---|---|
> > > |$3$|$30.17\pm0.61$|$31.47\pm0.74$|$\mathbf{35.03}\pm0.78$|
> > > |$4$|$56.63\pm0.37$|$56.80\pm0.37$|$\mathbf{60.47}\pm0.46$|
> > >
> > > **(C) Baseline = Ghazi et al. 2021 (no correction):**
> > > |$\epsilon$|Baseline|Full RT|Consensus-based RT|
> > > |---|---|---|---|
> > > |$3$|$23.60\pm0.92$|$29.23\pm1.03$|$\mathbf{36.30}\pm0.75$|
> > > |$4$|$48.25\pm0.05$|$52.10\pm0.10$|$\mathbf{57.40}\pm0.20$|
> > >
> > > As expected, forward and backward correction lead to better initial model performance (compared to no correction). The main thing to note however is that **consensus-based RT yields significant gains even with forward and backward correction**, consistent with our earlier results. Thus, consensus-based RT is a very effective post-processing step for improving learning with noisy labels. (It is worth noting that for $\epsilon=3$, consensus-based RT leads to similar accuracy with and without noise correction.)
> > >
> > > We hope you will take these extra results into consideration.

---

### Official Review · Reviewer_nHoC · 2025-03-13

**Overall Recommendation:** 4

**Summary:**

The paper gives a theoretical treatment on when learning with predicted hard label is beneficial than learning with original noisy label.

**Claims And Evidence:**

Yes, the claims were proved.

**Essential References Not Discussed:**

Essential references are included.

(Optional)
there can be some supplementary references that are related, see below.

**Experimental Designs Or Analyses:**

Yes.

**Methods And Evaluation Criteria:**

Overall makes sense to me. Though not quite sure why consider the "label DP" setup, seems to me a standard label noise setup suffices.

**Other Comments Or Suggestions:**

1) Notation: in label noise literature, people usually use $\tilde{y}$ to denote the noisy label, rather than $\hat{y}$. $\hat{y}$ is usually used to denote the predicted label by the classifier. This is a bit confusing.
2) line 197: "perfect separable setting", I don't think it's separable, because Gaussian has infinite support, therefore positive and negative classes are overlapping. Do you mean "Bayes decision boundary is linear"?
3) Eqn. 4.4: the notion $ P(sign(<x, \theta>) \neq y ) $ is overloaded, here it integrates $x$, while in Theorem 4.1, it is conditioned on $x$.

**Other Strengths And Weaknesses:**

Overall I enjoyed reading the paper.
The theoretical treatment of hard labels is new to me, and I think it is a good contribution to the literature.
Have some concerns at this point, I will be happy to read the authors' comments and re-assess my review.

My biggest theoretical concerns are:
1) The form of the classifier considered:
$$ \hat{\theta} = \frac{1}{n} \sum_{i=1}^n y_i x_i, $$
it does not correspond to any standard classifier.
(At first glance, I would expect ERM or logistic regression type.)

I look forward to see experimental setup that is more aligned with the theoretical setting:
1) a data simulation that corresponds exactly to the theoretical setting, e.g., a 2-dimensional mixture of two Gaussian (and use the exact form of the classifier in theory).
2) I think it is also possible to align CIFAR experiments with the theoretical setting, e.g., use a pretrained NN to extract the feature, then apply linear classifier on top of it (aka, "linear probing" in self-supervised learning).

These should provide stronger evidence to the theory.

**Questions For Authors:**

1) line 072, Figure 1: what's the classifier used to get the result? (linear, MLP, or the one in Eqn. 4.3/4.8?)
2) line 217: why need "u"? It seems a bit redundant and does not seem to play a key role.
3) Eqn. 4.3 & 4.8: my biggest concern, why the classifier takes the form of
$$ \hat{\theta} = \frac{1}{n} \sum_{i=1}^n y_i x_i, $$
it does not correspond to any standard classifier, and what's the multi-class version of it?

4) Theorem 4.6 (minimax lower bound): I am aware of three (non-parametric) lower bounds in [1-3], so I would like to know the position of this lower bound in the literature.
5) Theorem 4.6 also applies to the "retraining classifier" $\theta_1$ in Eqn. 4.8, therefore, predicted hard label do not provide a gain in terms of rate/sample complexity. Then the benefit of using the hard label is only in terms of the constants?


[1]  T Tony Cai and Hongji Wei. Transfer learning for nonparametric classification: Minimax rate and adaptive classifier. The Annals of Statistics, 49(1):100–128, 2021.

[2]  Hyungki Im and Paul Grigas. Binary classification with instance and label dependent label noise. arXiv preprint arXiv:2306.03402, 2023.

[3] Yilun Zhu, Jianxin Zhang, Aditya Gangrade, and Clayton Scott. Label Noise: Ignorance Is Bliss. In Advances in Neural Information Processing Systems, 2024

**Relation To Broader Scientific Literature:**

The benefit of using predicted hard label is has been studied empirically, the theoretical treatment is new.

**Theoretical Claims:**

I've skimmed through the proofs, but have not checked the details.

---

> ### Author Rebuttal · Authors · 2025-04-01
>
> Thanks for the review and great questions!
>
> **(A) Label DP setting.** We focused on this because it’s not clear how to apply existing noise-robust techniques on top of existing label DP mechanisms, while retraining is a simple post-processing step. For e.g., as mentioned in lines 365-366 right column, it’s not obvious how to apply forward correction to the second stage of Ghazi et al 2021.
>
> **(B) Form of classifier $\hat{\theta} =  \sum_i y_i x_i$.** This is a simplification to the least squares’ solution (LSS) obtained by removing the empirical covariance matrix’s inverse. The way to analyze the LSS would be to bound the deviation of the empirical covariance matrix from the population covariance matrix (which shrinks as $n \to \infty$), then analyze with the features pre-multiplied by covariance matrix’s inverse. This would just make the math more tedious w/o adding any meaningful insights. Also, as we wrote around eq. 4.3, our classifier is similar to kernel methods with the inner product kernel & it has been used in the highly-cited work of Carmon et al 2019.
>
> **(C) Experimental setup more aligned with theory setting**
>
> 1. Setting of Fig. 1 corresponds to our theory setting; see Appendix A.
>
> 2. We did linear probing (LP) for CIFAR-100 with features extracted from a ResNet-50 pretrained on ImageNet. The setup is similar to our full fine-tuning (FT) experiments; we omit details here due to lack of space. Results:
>
> |$\epsilon$|Baseline|Full RT|Consensus-based RT|
> |---|---|---|---|
> |$3$|$55.26\pm0.19$|$60.97\pm0.21$|$\mathbf{63.37}\pm0.26$|
> |$4$|$64.83\pm0.39$|$66.67\pm0.33$|$\mathbf{67.83}\pm0.37$|
>
> *So even here RT (especially, consensus RT) yields good gains*. Note that LP performs much better than full FT; this is often the case when training with noise due to less overfitting with LP.
>
> **(D) Line 197: perfect separable setting.** *Our modified GMM setting* (eq 4.1) is separable. As discussed in lines 172-176 right column, $\theta^{*} = \mu$ separates the data perfectly.
>
> We’ll fix/clarify notational ambiguities.
>
> **(E) Questions for Authors**
>
> **1.** The ones in eqs. 4.3 & 4.8. Also see Appendix A.
>
> **2.** We introduced $u$ so that the margin of a data point $x$ along $\mu$ is not the same. As explained in lines 172-175 right column, $|<x, \mu>| = (1+u)||\mu||^2 \geq ||\mu||^2$. If there is no $u$, all the points would have the same margin. We agree that from the analysis perspective, $u$ is not very important.
>
> **3.** For the binary case, see (B) above. Even in the multi-class case, something like this has been studied in reference [A] (see Section 2.2). Specifically, for $C$ classes, the labels $y_i$’s will be $C$-dimensional one-hot vectors, the ground truth $\Theta^{\ast}$ will be a $C \times d$ matrix (features $x_i$’s are still $d$-dimensional, but $y_i$’s need to be defined appropriately in terms of $\Theta^{\ast}$ & $x_i$’s) and our predictor $\hat{\Theta} = \frac{1}{n} \sum_i y_i x_i^T$ will also be in $\mathbb{R}^{C \times d}$.
>
> **4.** *These lower bounds are in much more general settings than ours and so they are weaker than ours*. In [1], our setting corresponds to $n_p = 0, n_q = n$. Per Definition 2 of [1], $\beta \leq 1$ and as per the paragraph after Remark 3, $\alpha \beta \leq d$. Now as per Thm 3.2, the lower bound on the error is effectively $n^{-O(\frac{1+\alpha}{d})}$. So when $\alpha \ll d$, this lower bound yields a much worse sample complexity than our result in Thm 4.6. In [2], the lower bound on the error (Thm 2) doesn’t reduce with $n$, so even if there are infinite samples, we can’t get 0 error in the worst case. As for [3], the lower bound on the error (Thm 1) also has a non-diminishing term depending on $\epsilon$. In the special case of $\epsilon = 0$ (or $\epsilon$ being small), there is a $n^{-1/2}$ dependence but no dependence on the dimension (or a related quantity). But their upper bound in Thm 2 does have a dependence on a VC dimension-like quantity as expected, so their lower bound is probably loose w.r.t. dimension.
>
> **5.** Yes, Thm. 4.6 also applies to the retraining (RT) classifier. If you see Remark 4.10, the min. # of samples $n$ needed for RT to be better is **more than** $d/(1-2p)^2$, i.e., the lower bound of Thm. 4.6. And as discussed after Remark 4.10, this requirement on $n$ is probably tight (modulo log factors) because we can only hope the RT classifier to be better if the accuracy of the labels with which it is trained – namely, the initial model’s predicted labels – is more than $(1-p)$ (= accuracy of the noisy labels with which the initial model is trained); this requires at least $d/(1-2p)^2$ samples (per Thm. 4.6). So yes, RT can’t improve the sample complexity beyond $d/(1-2p)^2$.
>
> We hope to have resolved your concerns and are happy to discuss further. If you’re satisfied, *we hope you will raise your score*!
>
> ---
>
> [A]: "Theoretical Insights Into Multiclass Classification: A High-dimensional Asymptotic View", Thrampoulidis et al., 2020

---

> > ### Comment · Reviewer_nHoC · 2025-04-04
> >
> > Thank you for the detailed response.
> >
> > It would be nice to see a more comprehensive discussion of the lower bound in the next version of the paper (can be a formalized version of the response).
> > Given the limited exploration of lower bounds in the existing label noise literature, I think it is a good addition.
> >
> > Regarding the debate on "linear separability," my understanding is that distributions are separable if and only if their supports do not overlap. However, this difference in perspective is minor, and I am comfortable moving forward despite differing views.
> >
> > Overall, the authors have adequately addressed my concerns, and I anticipate the next version of the manuscript will provide further clarity. I recommend acceptance (and have raised the score from 3 to 4).

---

> > > ### Author Response · Authors · 2025-04-04
> > >
> > > Thank you for raising your score! Yes, we will add a discussion on the lower bounds in the next version and we agree, it’ll be a good addition. Thanks for pointing out these papers! We’ll also clarify what we mean by separability and add the extra experiments.

---

### Decision · Program_Chairs · 2025-05-01

**Decision:**

Accept (poster)

**Comment:**

This is a case where author's response has resulted in two reviewers increasing their original scores so that overall paper score now is 4,4,3. The responses have addressed the concerns and the reviewers have increased the scores with the condition that the authors will include the additional results (e.g. discussions and formalizing the lower bounds and additional experimental results) in the camera ready. I recommend acceptance.